# Spectral-Refiner: Accurate Fine-Tuning of Spatiotemporal Fourier Neural Operator for Turbulent Flows

**Shuhao Cao**[*]
School of Science and Engineering
University of Missouri-Kansas City

**Francesco Brarda**
Department of Mathematics
Emory University

**Rui Peng Li**
Center for Applied Scientific Computing
Lawrence Livermore National Laboratory

**Yuanzhe Xi**[*]
Department of Mathematics
Emory University

## Abstract

Recent advancements in operator-type neural networks have shown promising results in approximating the solutions of spatiotemporal Partial Differential Equations (PDEs). However, these neural networks often entail considerable training expenses, and may not always achieve the desired accuracy required in many scientific and engineering disciplines. In this paper, we propose a new learning framework to address these issues. A new spatiotemporal adaptation is proposed to generalize any Fourier Neural Operator (FNO) variant to learn maps between Bochner spaces, which can perform an arbitrary-length temporal super-resolution for the first time. To better exploit this capacity, a new paradigm is proposed to refine the commonly adopted end-to-end neural operator training and evaluations with the help from the wisdom from traditional numerical PDE theory and techniques. Specifically, in the learning problems for the turbulent flow modeled by the Navier-Stokes Equations (NSE), the proposed paradigm trains an FNO only for a few epochs. Then, only the newly proposed spatiotemporal spectral convolution layer is fine-tuned without the frequency truncation. The spectral fine-tuning loss function uses a negative Sobolev norm for the first time in operator learning, defined through a reliable functional-type a posteriori error estimator whose evaluation is exact thanks to the Parseval identity. Moreover, unlike the difficult nonconvex optimization problems in the end-to-end training, this fine-tuning loss is convex. Numerical experiments on commonly used NSE benchmarks demonstrate significant improvements in both computational efficiency and accuracy, compared to end-to-end evaluation and traditional numerical PDE solvers under certain conditions. The source code is publicly available at https://github.com/scaomath/torch-cfd.

## 1 Introduction

Recently, Deep Learning (DL) pipelines have proven particularly effective in addressing problems formulated by Partial Differential Equations (PDEs). In this paper, we study the problem of learning Neural Operators (NOs) between infinite-dimensional function spaces (Boullé & Townsend, 2023; Kovachki et al., 2023; Azizzadenesheli et al., 2024; de Hoop et al., 2022; 2023). Specifically, the problem in consideration is for the Navier-Stokes Equations (NSE) in the turbulent regime ($Re = \mathcal{O}(10^3)$ to $\mathcal{O}(10^4)$). In computation, the difficulties of solving NSE in this regime roots from its "stiffness" attributed to the nonlinear convection with a nearly singular viscous diffusion, as well as the spatiotemporal nature of being highly transient. For this problem, we propose to synergize operator learning with classical numerical PDE methods, complementing one's drawbacks and limitations with the other's strengths.

---

[*]Corresponding to scao@umkc.edu or yuanzhe.xi@emory.edu

**Compromises and drawbacks of traditional numerical methods.** To solve NSE, traditional time stepping schemes include Adams-Bashforth/-Moulton or Runge-Kutta (RK) families, e.g., see Canuto et al. (2007, Appendix D), Karniadakis et al. (1991). If one opts to use an explicit scheme, or there exists a certain portion of the forcing terms (e.g., the convection term in NSE) computed via explicit schemes, then small time step sizes must be used: $\Delta t \sim \mathcal{O}\left((\Delta x)^\alpha\right), \alpha \geq 1$ (Rannacher, 2000, Chapter 4). This necessity is often referred to as the "stiffness" of the PDE, and the threshold or constraint of the time step is called the Courant-Friedrichs-Lewy (CFL) condition, e.g., see Johnston & Liu (2004); Wang (2012); He & Sun (2007). CFL poses a sufficient condition on the step size for "stability", and this requires the step size usually much smaller than the requirement for "consistency". Here, this temporal consistency usually refers to a first-order optimal local truncation error, e.g., how the original Butcher tableau is derived for RK (Butcher, 1965). The CFL puts a threshold for ***all*** explicit time-marching schemes on how fast the local errors in different frequencies can propagate, and thus prevent error accumulation "pollutes" the approximation globally to ensure stability. This means that, for any time-marching schemes, finer mesh (better spatial consistency) requires the time steps to be much smaller to prevent errors from traveling to neighboring nodes and elements, which greatly increases the computational cost.

**Spatiotemporal operator learning.** Among the end-to-end operator learning for NSE, the common approach is the so-called autoregressive "roll-out". During rolling-out, several snapshots of solutions are concatenated as ***channels*** in the input tensor to the neural operator, which outputs an approximated solution at the subsequent time step. This procedure can be repeated recurrently until reaching the model's stability capacity. In contrast to the traditional time marching solvers whose step sizes are restricted by the CFL condition, roll-out can withstand a much bigger time step size. NOs used in roll-out approach include the original FNO2d (Li et al., 2021), and those in Li et al. (2022); Brandstetter et al. (2023); Gupta & Brandstetter (2023); Fonseca et al. (2023); Zheng et al. (2024). However, for an end-to-end operator learner, the roll-out approach faces the same dilemma of super-resolution capacity in the temporal dimension as a traditional solver: finer time steps cost prohibitively more, while larger time steps does not guarantee stability. To solve this problem, we turn to the prevalent functional analytic framework for studying the convergence and stability of solution trajectories for NSE: the theory of Bochner spaces, e.g., Aubin-Lions lemma (Lions & Magenes, 2012, Chapter 3), (Temam, 1995, Chapter 2), (Evans, 2022, Chapter 7). Inspired by this theoretical insight, we propose a **S**patio**T**emporal adaption for all **F**ourier **N**eural **O**perator (ST-FNO) variants. ST-FNO now can perform arbitrary zero-shot super-resolution not only spatial dimensions, but also allows the temporal dimension to vary for the first time. For a prototype spatiotemporal operator learner FNO3d in Li et al. (2021), it learns a map between a fixed number of spatial "snapshots" (product spaces). The newly proposed ST-FNO directly learns a map between Bochner spaces $L^2(\mathcal{T}_0; \mathcal{V})$ to $L^2(\mathcal{T}; \mathcal{V})$ on non-overlapping time intervals $\mathcal{T}_0$ and $\mathcal{T}$. This makes the model become "trajectory-to-trajectory", where the initial trajectory is the input of an NO to obtain the output evaluation as an approximation to the subsequent trajectory of the solution. Here $\mathcal{V}$ denotes the spatial Hilbert space in which snapshots of the solution at a specific time reside, and for a detailed notation list, we refer the readers to Appendix A.

**Limits and the lack of accuracy for neural operators.** The NO approach has the potential to bypass various difficulties and compromises of traditional schemes of numerical PDEs. However, in all end-to-end operator learning benchmarks of NSE, even the state-of-the-art NOs still fall short in achieving high-accuracy solutions. For example, an end-to-end roll-out approach suffers from error accumulation experimentally, e.g., see Li et al. (2022, Figure 9). To our best knowledge, in terms of the relative difference with the ground truth under the Bochner norm, no NO-only-based operator learning approach can break the barrier of an even 2-digit accuracy. Moreover, to our best knowledge, NO-only approaches have no theoretical stability estimate, such as the error propagation operator is a contraction. Recently, a remarkable advancement called PDE-Refiner (Lippe et al., 2023) learns an extra error correction NO under the Denoising Diffusion framework (Ho et al., 2020). For a single instance, it can get a stable long roll-out, and achieve for the first time $\mathcal{O}(10^{-8})$ relative difference with the ground truth after a single time marching step, and $\mathcal{O}(10^{-6})$ in the Bochner norm. Nevertheless, for all models above and their learning frameworks, the optimization is to minimize the difference between the outputs from the NO, namely $u_\mathcal{N}$, and the ground truth $u_\mathcal{S}$, generated by a traditional numerical PDE solver. The difference with the true solution $u$ under a certain norm is not directly optimized, as the analytical expression of $u$ is unknown most of the time in real-life applications. For difficult PDEs such as the NSE, in a linear time-stepping scheme, the ground truth $u_\mathcal{S}$ may on its own only have 3-digit accuracy in the Bochner norm. Note that this may already

be of optimal order $\mathcal{O}(\Delta t)$ by convergence estimates (Heywood & Rannacher, 1986). Therefore, minimizing the difference between $u_{\mathcal{N}}$ and $u_{\mathcal{S}}$ becomes fruitless if the numerical approximation (ground truth) $u_{\mathcal{S}}$ is not accurate at the first place, as unnecessary computational resources may have been spent to get closer to $u_{\mathcal{S}}$.

**New hybrid scheme.** To address these dilemmas for NSE's approximation, we take an alternative hybrid learning paradigm inspired by traditional numerical methods. Unlike the arduous training of running hundreds of epochs, the newly proposed ST-FNO is trained for only a few epochs (e.g., 10), concluding with the freezing of most model parameters. Then, the newly proposed spatiotemporal spectral convolution, as the *last* layer, in ST-FNO is fine-tuned with the help of traditional solvers. Note that SF-FNO's temporal arbitrary-length inference capacity is attributed to this last layer. During fine-tuning, this layer is relieved from the common frequency truncation FNO has. A traditional solver with a single time step is used to obtain highly accurate approximations for extra field variables to conform with the physics. These extra fields help the evaluation of a new loss function in fine-tuning, a functional-type *a posteriori* error estimate measured under a *negative* Sobolev norm, which is equivalent to the variational form-associated norm for the NSE. Note that there shall be no training of extra error-correcting NOs (Dresdner et al., 2023; Lippe et al., 2023). Moreover, unlike the same nonconvex optimization problem that the extra error-correcting NOs try to tackle as the one in the original end-to-end training, our fine-tuning optimization problem is *convex*, and allows us to achieve high accuracy with a fractional of computational resources when compared with other fine-tuning approaches that optimize the whole model with a physics-informed loss.

**Fine-tuning using functional-type a posteriori error estimation.** To seek highly accurate NO approximated solutions that are closer to the true solution directly, we turn to the *a posteriori* error estimation techniques. This technique allows computing the error without knowing the true PDE solutions, and has been widely studied for Galerkin-type methods, such as finite element (Ainsworth & Oden, 1993; 1997; Oden et al., 1994), as well as for parameterized PDEs (Hesthaven et al., 2016; Patera et al., 2007). Among all types of *a posteriori* error estimation techniques, functional-type *a posteriori* estimator (Repin, 2008) views the error as a functional on the test Sobolev spaces and evaluates accurate representations with the help of an extra dual variable (Ern & Vohralík, 2010). In our hybrid approach, inspired by this, we combine the strengths of NOs and traditional numerical solvers. Using the newly proposed spatiotemporal discretization-invariant NO, we propose to use a negative Sobolev norm in the frequency domain as a functional-type *a posteriori* error estimation as the fine-tuning loss. Traditionally, the *a posteriori* error estimation is used to refine local basis, yet this constraint on the purpose of the method leads to inaccurate localized representations for the $H^{-1}$ error functional. Our new approach is not attached to the local refining requirement (Bonito et al., 2024). As a result, the negative Sobolev norm used in the new loss is handily defined through the Gelfand triple in the frequency domain which is global. Our method needs no extra dual variable reconstruction problem as in the traditional methods (Ern & Vohralík, 2010). Unlike commonly adopted physics-informed operator learning (Li et al., 2024d), NO prediction for other field variables are not needed either. The extra field variables for computing the error are recovered through an auto-differentiable numerical solver with the NO output of the primal variable (vorticity or velocity). Overall, this practice leads to "refining" the learnable set of the global spectral basis.

For a more detailed review of operator learning and further motivation discussion with a much higher degree of mathematical rigor, we refer the readers to Appendix B.

**Main contributions.** The main contributions of this work are summarized as follows. For the difference between the common roll-out approach versus direct spatiotemporal learning between Bochner spaces, we refer the readers to Figure 1.

- **Spatiotemporal Fourier Neural Operator.** We design the first spatiotemporal adaption technique for all FNO variants (ST-FNO) to enable them to learn maps between Bochner spaces.
- **New hybrid operator learning paradigm.** We propose to train and evaluate ST-FNOs using a new strategy, which has significantly improved over the existing methods in speed and accuracy. Only a few epochs of training combined with a spectral-refining fine-tuner yield highly accurate approximations. Extra field variables are obtained by auto-differentiable single-step (RK4) or multistep (Adam-Bashforth) solvers, the input of which are neural predictions. The auto-differentiability of the solver makes it possible for the fine-tuning to optimize the parameters in FNO.
- **Functional a posteriori error estimation.** Leveraging the spectral structure of the new SpatioTemporal layer in ST-FNO and Parseval identity, the fine-tuning utilizes for the first time the negative

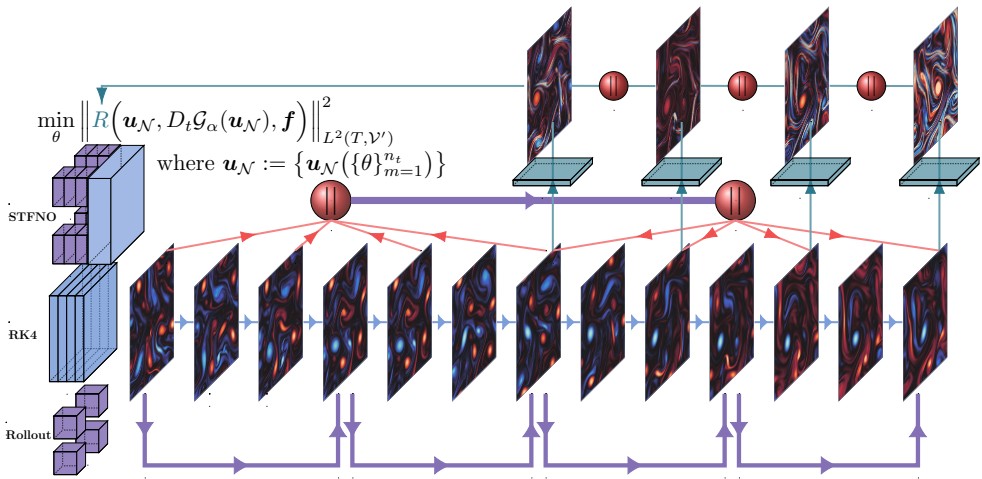

$$\min_\theta \left\| R\Big(\boldsymbol{u}_\mathcal{N}, D_t \mathcal{G}_\alpha(\boldsymbol{u}_\mathcal{N}), \boldsymbol{f}\Big) \right\|^2_{L^2(T, \mathcal{V}')}$$

where $\boldsymbol{u}_\mathcal{N} := \{\boldsymbol{u}_\mathcal{N}(\{\theta\}^{n_t}_{m=1})\}$

STFNO

RK4

Rollout

Figure 1: Schematic differences between approaches. 4th-order Runge-Kutta (RK4): small time steps bounded by the CFL condition, de-aliasing filter needed; autoregressive NO rolling-out: using previous evaluation as input repetitively, large time steps, no stability guarantees; Spatiotemporal FNO (ST-FNO) with hybrid fine-tuning: large time steps, yielding arbitrary-length temporal prediction in a single evaluation, parallel-in-time optimization for spectral fine-tuning.

Sobolev norm (functional norm) as loss via *a posteriori* error estimation. This procedure does not require any ground truth data (e.g., numerical solution generated by traditional numerical solvers), nor the knowledge of the true solution(s) to the PDE. This new loss is proven to be reliable in theory, in the meantime much more efficient in the ablation experiments.

- **Experimental results.** We created a native PyTorch port of Google's Computational Fluid Dynamics written in JAX (Kochkov et al., 2021; Dresdner et al., 2023) with enhanced functionality for tensor operations such as the facilitation of fine-tuning for the latent fields, publicly available at `https://github.com/scaomath/torch-cfd`, with scripts to replicate the experiments as well as the data generation. The data are available at `https://huggingface.co/datasets/scaomath/navier-stokes-dataset`.

## 2 SPATIOTEMPORAL OPERATOR LEARNING FOR NAVIER-STOKES EQUATIONS

Both drawbacks and advantages of traditional numerical methods and NO-based methods in Section 1 play vital roles in shaping our study. We first briefly discuss the spatiotemporal operator learning problem on Bochner spaces associated with NSE. Then, we detail how to modify a generic FNO-based neural architecture to obtain an operator learner between Bochner spaces.

### 2.1 SPATIOTEMPORAL OPERATOR LEARNING PROBLEM FOR NSE

For 2D NSE, in the the velocity-pressure (V-P) formulation (2.2), the velocity field $\boldsymbol{u}(t, \boldsymbol{x}) : [0, T] \times \Omega \to \mathbb{R}^2$ is seen as an element in the Bochner space $L^p([0, T], \mathcal{V})$ where $\mathcal{V}$ is a spatial Sobolev space in which each snapshot at $t$ of the solution $\boldsymbol{u}(t, \cdot)$ resides. As in Li et al. (2021), we also consider the vorticity-streamfunction (V-S) formulation (2.1) with periodic boundary condition (PBC). In $[0, T] \times \Omega$, the scalar-valued vorticity $\omega := \nabla \times \boldsymbol{u}$, and streamfunction $\psi$ satisfies $\boldsymbol{u} = \mathbf{rot}\,\psi$. These two formulations read

(Vorticity-Streamfunction)      $\partial_t \omega + \mathbf{rot}\,\psi \cdot \nabla \omega - \nu \Delta \omega = \nabla \times \boldsymbol{f}, \quad \omega + \Delta \psi = 0.$    (2.1)

(Velocity-Pressure)      $\partial_t \boldsymbol{u} + (\boldsymbol{u} \cdot \nabla)\boldsymbol{u} - \nu \Delta \boldsymbol{u} + \nabla p = \boldsymbol{f}, \quad \nabla \cdot \boldsymbol{u} = 0.$    (2.2)

For all analyses in line with the Hilbertian framework, $\mathcal{V} = H^1(\mathbb{T}^2)$ for vorticity and $\boldsymbol{H}^1(\mathbb{T}^2)$ for velocity, where $\mathbb{T}^2$ denotes the unit torus, i.e., $\Omega = (0, 1)^2$ with a component-wise PBC. For a fixed forcing function in $\mathcal{V}'$, the initial condition is either $\omega(0, \boldsymbol{x}) = \omega_0(\boldsymbol{x})$ or $\boldsymbol{u}(0, \cdot) = \mathbf{rot}\big((-\Delta)^{-1}\omega_0\big)$. Here $\omega_0$ is drawn from a compactly supported probability measure $\mu$, in which the compactness corresponds to certain power/enstrophy spectrum decay law to produce isotropic turbulence (McWilliams,

1984), and we refer the reader to Appendix C for details. Then, we can consider the following map $\overline{G}^\mu : \mathcal{X}_\mu \Subset \Pi_{i=1}^\ell \mathcal{V} \to \Pi_{i=1}^{n_t} \mathcal{V}$:

$$\overline{G}^\mu : \mathbf{a} := \left[\omega(t_1, \cdot), \dots, \omega(t_\ell, \cdot)\right] \mapsto \mathbf{u} := \left[\omega(t_{\ell+1}, \cdot), \dots, \omega(t_{\ell+n_t}, \cdot)\right], \qquad (2.3)$$

where $t_{\ell+n_t} \leq T$. The input and the output in the training data for $\overline{G}^\mu$ are snapshots obtained from a solution trajectory with the same initial condition $\omega_0$ that is drawn from certain compactly supported probability measure $\mu$. The operator learning task for NSE using a prototype spatiotemporal operator learner, e.g., FNO3d (Li et al., 2021), is to learn this $\overline{G}$ between a fixed number of Cartesian products of spatial Sobolev spaces. Specifically, $\overline{G}^\mu : \mathcal{X}_\mu \to \mathcal{Y}$ maps elements in $\mathcal{X}_\mu \Subset \Pi_{i=1}^\ell \mathcal{V}$ to elements in $\mathcal{Y} = \Pi_{i=1}^{n_t} \mathcal{V}$. In the task for this prototype "snapshots learner", the number of snapshots $\ell \in \mathbb{Z}^+$ is **fixed**. As such, $\mathcal{X}_\mu$ and $\mathcal{Y}$ represent spaces of two non-overlapping temporal segments of solution snapshots. Based on these snapshots' discretized approximations from data, these snapshots learners learn an operator $\overline{G}_\theta : \mathcal{X} \to \mathcal{Y}$, where the number of parameters in $\overline{G}_\theta$ is independent of the spatial discretization size, yet **does depend on the number of snapshots** $\ell$. This dependence in the setting of this task makes it not "trajectory-to-trajectory".

In this study, the learning aims to recast (2.3) to a trajectory-to-trajectory operator learning problem, conforming to the Hilbertian formulation of NSE using Bochner spaces. Using the fact that the weak solutions to (2.1) and (2.2) at a given time interval $\mathcal{T}$ is in $L^2(\mathcal{T}; \mathcal{V})$, the operator to be learned is

$$G^\mu : L^2(t_1, t_\ell; \mathcal{V}) \to L^2(t_{\ell+1}, t_{\ell+n_t}; \mathcal{V}) \text{ for } \mathcal{V} := H^1(\mathbb{T}^2) \text{ or } \{\boldsymbol{v} \in \boldsymbol{H}^1(\mathbb{T}^2) : \nabla \cdot \boldsymbol{v} = 0\}. \quad (2.4)$$

In what follows, we shall present how to modify any common FNO variant to become a Bochner space operator learner that can learn maps from arbitrary-sized discretization in $\mathbb{R}^{\ell \times n \times n \times d}$ to $\mathbb{R}^{n_t \times n \times n \times d}$ ($d = 1$ in V-S; $d = 2$ in V-P). This operator can be trained by a standard end-to-end supervised learning pipeline using lower-resolution data, and for inference, the newly proposed spatiotemporal trajectory-to-trajectory learner, $\ell, n_t, n$ can all be of variable sizes. In contrast, the snapshot learner FNO3d predicts the subsequent $n_t$ snapshots with the first $\ell$ snapshots as input, with both $n_t$ and $\ell$ fixed, and only the spatial resolution $n$ can be much higher than the input-output pairs used in the training. Hereafter we omit the $d$ dimension if no ambiguity arises.

## 2.2 SPATIOTEMPORAL ADAPTATION OF FOURIER NEURAL OPERATORS

For any FNO variant, such as FNO3d (Li et al., 2021) or Factorized FNO (Tran et al., 2023), with the following meta-architecture: $G_\theta := Q \circ \sigma_L \circ K_{L-1} \circ \cdots \circ \sigma_1 \circ K_0 \circ P$, where $P$ is a lifting operator, $Q$ is a projection operator that does a pointwise reduction in the channel dimension, $\sigma_j$ can be a pointwise universal approximator or simply chosen as a nonlinearity. $K_j := K_{\phi_j}$ is the parametrized spectral convolution that uses a spatiotemporal 3D FFT. During training, the operator to be learned is restricted to finite-dimensional subspaces $\mathcal{X} \supset \mathcal{X}_n \to \mathcal{Y}_n \subset \mathcal{X}$, in the sense that the continuous spatial Sobolev space $\mathcal{V}$ in the product spaces is replaced by a finite-dimensional subspace $\mathcal{V} \supset \mathcal{S} \simeq \mathbb{R}^{n \times n}$ with continuous embeddings $\{\mathbf{a}_\mathcal{S} \in \mathbb{R}^{\ell \times n \times n}\} \hookrightarrow \mathcal{X}$, and $\{\mathbf{u}_\mathcal{S} \in \mathbb{R}^{n_t \times n \times n}\} \hookrightarrow \mathcal{Y}$. The positional encodings $\mathbf{p} := (t_i, \boldsymbol{x}_j)_{i=1}^\ell \in \mathbb{R}^{3 \times \ell \times n \times n}$ represents the underlying spatiotemporal grid, and is concatenated to $\mathbf{a}_\mathcal{S}$ before feeding to $P$.

For spatiotemporal problems, all FNO variants novelly exploit a convenient architectural advantage of operator-valued NNs: the input temporal dimension $\ell$ is treated as the channel dimension of an image, thus enabling channel mixing as a learnable temporal extrapolation. However, this neat trick coincidentally makes the lifting operator the biggest hurdle for $G_\theta$ to become a trajectory-to-trajectory operator learner between Bochner spaces, since the channel dimension of $P$ must be hard-coded[1] and thus depends on the input pair's time steps.

In what follows, we use FNO3d as an example to present three new modifications to FNO3d to become a trajectory-to-trajectory learner ST-FNO3d, that is, to act as an operator that maps an arbitrary-time-step input to an arbitrary-time-step output. We also note that this modification applies to any FNO variant that predicts spatial and temporal dimensions at the same time. These changes are so simple that the trajectory-to-trajectory modification can serve as drop-in replacements for their snapshot learner counterparts, when the temporal input dimensions are fixed in a task. For the schematic difference, please refer to Figure 2, Figure 3, and Appendix D.2 for more comments.

---

[1] Line 97 in the original FNO3d source code `fourier_3d.py` (link provided for an unaltered fork of the `master` branch commit `de514f2` in the original FNO GitHub repository).

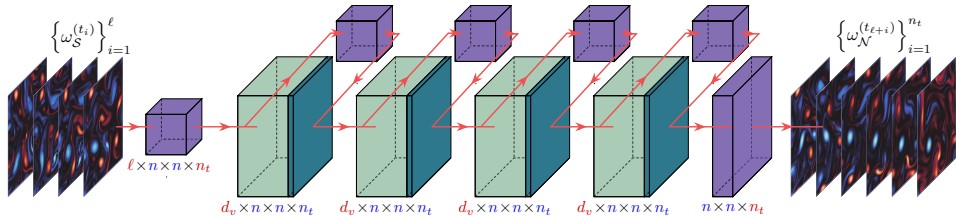

Figure 2: The FNO3d in Li et al. (2021) is a snapshot learner. ▮: spectral convolution layer ; ▮: pointwise nn.Conv3d that works as channel expansion/reduction; ▮: pointwise nonlinearity. The TikZ source code to produce this figure is modified from the examples in Iqbal (2018).

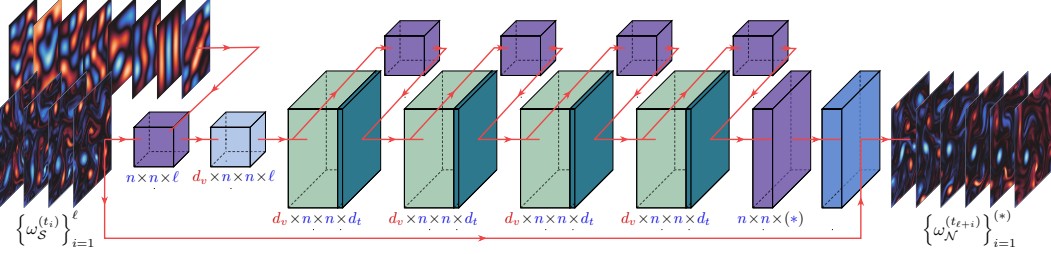

Figure 3: The Spatiotemporal-adapted FNO3d (ST-FNO3d) is now a trajectory-to-trajectory learner. ▢: layer normalization to replace a hard-coded global normalization. Combined with channel mixing, the first spectral convolution layer serves as a time-depth-wise separable (global) convolution, after which the time dimension is shrank to a fixed "latent" time dimension through iFFT's resampling. ▮: the spatiotemporal spectral convolution layer as the final layer is fine-tuned after the training phase.

**Modifications to 3D FFT.** The first change is in the spatiotemporal FFT from an analytical point of view. Using the spectral convolution $K(\cdot)$ in FNO3d as an example, with a slight abuse of notation, denote the latent dimension (width) by $d_v$, let the vector-valued latent representation with $d_v$ channels be continuously embedded into $\Pi_{i=1}^{n_t}\mathcal{V}$ by $\mathbf{v}_{\mathcal{S}} \hookrightarrow \mathbf{v}$ in each channel, and denote $\Lambda := [t_{\ell+1}, \ldots, t_{\ell+n_t}]$, then $K(\cdot)$ is "semi-discrete" in a sense as follows

$$(K\mathbf{v})(t,\boldsymbol{x}) := (W\mathbf{v})(t,\boldsymbol{x}) + \sum_{s\in\Lambda}\int_\Omega \kappa\big((t,\boldsymbol{x}),(s,\boldsymbol{y})\big)\mathbf{v}(s,\boldsymbol{y})\mathrm{d}m(\boldsymbol{y}), \tag{2.5}$$

where $t$ takes values only in a discrete set $\Lambda$, and $\boldsymbol{x} \in \Omega$. Here $W \in \mathbb{R}^{(d_v+1)\times d_v}$ is a pointwise-applied affine linear operator, $\kappa \in C\big((\Lambda \times \Omega) \times (\Lambda \times \Omega); \mathbb{R}^{d_v \times d_v}\big)$, and $m$ denotes an approximation to the Lebesgue measure on $\Omega$. Evolving $K$ into a spatiotemporal spectral convolution acting on Bochner space is straightforward. We change (2.5) slightly by adopting of an atom-like measure $\delta(\cdot)$ in the temporal dimension, which then generalizes to the variable-length temporal discretization for any $(t,\boldsymbol{x}) \in (a,b) \times \Omega$, i.e., one can obtain arbitrarily many snapshots on the interval of interest

$$(\widetilde{K}(\mathbf{v}))(t,\boldsymbol{x}) := (W\mathbf{v})(t,\boldsymbol{x}) + \int_a^b \int_\Omega \kappa\big((t,\boldsymbol{x}),(s,\boldsymbol{y})\big)\mathbf{v}(s,\boldsymbol{y})\,\mathrm{d}m(\boldsymbol{y})\mathrm{d}\delta(s). \tag{2.6}$$

During training using the discretized data, each hidden layer is a map from $\mathbb{R}^{d_v\times d_t\times n\times n} \to \mathbb{R}^{d_v\times d_t\times n\times n}$, where $d_t$ is a "latent" dimension of time steps that is chosen $\leq n_t$. The layerwise propagation mechanics remains the same: $\widetilde{K}_\phi \boldsymbol{v} = W\boldsymbol{v} + \mathcal{F}^{-1}\left(R_\phi \cdot (\mathcal{F}\boldsymbol{v})\right)$, where $\mathcal{F}$ and $\mathcal{F}^{-1}$ denote the spatiotemporal Fourier transform and its inverse, respectively. The global spatiotemporal interaction characterized by the kernel is truncated in terms of modes in the frequency domain at $(\tau_{\max}, k_{\max})$, such that $\{(\tau, \boldsymbol{k}) \in \mathbb{Z}^3 : |\tau| \leq \tau_{\max}, |\boldsymbol{k}_j| \leq k_{\max}, j = 1, 2\}$ are kept. $R_\phi$ is parametrized as a $\mathbb{C}^{\tau_{\max}\times k_{\max}\times k_{\max}\times d_v\times d_v}$-tensor. Here $(\tau, \boldsymbol{k})$ denotes the coordinate in the spatiotemporal Fourier domain. Note that the integral represented by matrix product with $\mathcal{F}\kappa(\cdot,(s,\boldsymbol{y}))$ can then be viewed as residing in the continuous space as the Fourier basis (3.1) can be evaluated at arbitrary point.

**Modifications to the lifting operator $P$.** The other major hurdle for FNO3d to become trajectory-to-trajectory is that a global normalization is applied with a hard-coded temporal dimension using training

data.[2] In ST-FNO3d's modification, a periodic padding along the temporal dimension is applied to the input, and then a time-depth separable convolution layer $\mathcal{I}$ with variable time steps is used to map an arbitrary number of snapshots to a fixed number of channels $d_v$ with a fixed latent time steps $d_t$. The spatiotemporal positional encodings $\widetilde{\mathbf{p}}_{\mathcal{S}} := W_p \mathbf{p}$, with $W_p$ a learnable random projection with periodically padded $\mathbf{p}$ as input. This makes $\widetilde{\mathbf{p}}_{\mathcal{S}}$ match the latent fields' dimensions such that they can be concatenated. The latent fields, concatenated with $\widetilde{\mathbf{p}}_{\mathcal{S}}$, are then normalized by a learnable layer normalization layer $\mathrm{Ln}(\cdot)$, instead of a global fixed pointwise normalizer for the raw data in FNO3d. Finally, in ST-FNO3d, the lifting operator becomes $\widetilde{P} := \mathrm{Ln}(\widetilde{\mathbf{p}}_{\mathcal{S}} + \mathcal{I}(\cdot)) : \mathbb{R}^{\ell \times n \times n} \to \mathbb{R}^{d_v \times d_t \times n \times n}$.

**Modifications to the projection operator $Q$.** In FNO3d, the original out projection operator $Q$ maps a tensor in $\mathbb{R}^{n_t \times d_v \times n \times n}$ to another in $\mathbb{R}^{n_t \times n \times n}$. In the spatiotemporal-adapted FNO $\widetilde{Q} : \mathbb{R}^{d_v \times d_t \times n \times n} \to \mathbb{R}^{n_t \times n \times n}$, which maps the last latent representation to match the dimension of the output $\mathbf{u}_{\mathcal{S}}$. In $\widetilde{Q}$, the key for an arbitrary-length inference is that we compose another spectral convolution $K_{\mathcal{S}}$ after channel reduction. $K_{\mathcal{S}}$ can be thought of as a post-processing layer (also as the **only layer** to be fine-tuned). It takes advantage of the FFT/iFFT's natural super-resolution capacity, especially in the temporal dimension, by zero-padding the latent temporal step dimension ($d_t$) to given arbitrary output time steps. For the necessity of this padding, we refer the reader to Figure 7 for an illustrative example. For the V-P formulation, to impose the divergence-free condition for, $S$ is implemented as an optional non-learnable Helmholtz decomposition layer in $\widetilde{Q}$.

## 3 NEW HYBRID PARADIGM FOR SPATIOTEMPORAL OPERATOR LEARNING

Built upon a reasonably accurate approximation, the fine-tuning of ST-FNO is proposed. It is able to yield accuracy on par with traditional numerical methods on the same time horizon, and computational resources used are comparable to the evaluation of NOs. Taking advantage of the efficient ST-FNO zero-shot arbitrary-length temporal inference, this new approach does not need thousands of marching steps like traditional methods. Meanwhile, it borrows the wisdom from traditional Galerkin methods to improve the accuracy (consistency) of the NO approach, liberating the scheme from trade-offs that the traditional methods must make to ensure stability and convergence.

### 3.1 A POSTERIORI ERROR ESTIMATION USING A FUNCTIONAL NORM

We shall present the proposed fine-tuning using the V-P formulation in subsequent subsections. Denote an ST-FNO evaluation at a specific $t_m \in \mathcal{T}_{n_t} := \{t_{\ell+1}, \cdots, t_{\ell+n_t}\}$ in the output time interval as $\mathbf{u}_{\mathcal{N}}^{(m)}$. The construction of $\widetilde{Q}$ in ST-FNO renders $\mathbf{u}_{\mathcal{N}}^{(m)} \in \mathcal{W}$, where $\mathcal{W}$ is the divergence-free subspace of $\mathcal{S}|_{t=t_m} \times \mathcal{S}|_{t=t_m} \subset \mathcal{V} := \boldsymbol{H}^1(\mathbb{T}^2)$, where

$$\mathcal{S} := \mathrm{span}\left\{ \mathfrak{Re}\left(e^{i\tau_m t} e^{i\boldsymbol{k}\cdot\boldsymbol{x}}\right) : \ -n/2 \le k_j \le n/2 - 1, -n_t/2 \le m \le n_t/2 - 1 \right\}/\mathbb{R}, \quad (3.1)$$

where $\boldsymbol{k} := 2\pi(k_j)_{j=1,2}$ and $\tau_m := 2\pi m/(T - t_l)$. Then, a temporal continuous approximation $\mathbf{u}_{\mathcal{N}} := \mathbf{u}_{\mathcal{N}}(t, \cdot)$ can be naturally defined by allowing $t$ vary continuously on $\mathcal{T} := [t_{\ell+1} - t_p, T + t_p]$ thanks to the spectral basis of $\mathcal{S}$, where $t_p$ is the temporal periodic padding in Section 2.2. Define residual functional $R(\mathbf{u}_{\mathcal{N}}) \in L^2(\mathcal{T}; \mathcal{V}')$: at $t \in \mathcal{T}$ and for $\boldsymbol{v} \in \mathcal{V}$

$$R(\mathbf{u}_{\mathcal{N}}) := \boldsymbol{f} - \partial_t \mathbf{u}_{\mathcal{N}} - (\mathbf{u}_{\mathcal{N}} \cdot \nabla)\mathbf{u}_{\mathcal{N}} + \nu \Delta \mathbf{u}_{\mathcal{N}}, \ \text{ and } \ R(\mathbf{u}_{\mathcal{N}})(\boldsymbol{v}) := \langle R(\mathbf{u}_{\mathcal{N}}), \boldsymbol{v} \rangle. \quad (3.2)$$

$R(\mathbf{u}_{\mathcal{N}})$ measures how PDE (2.2) is violated by the finite-dimensional approximation $\mathbf{u}_{\mathcal{N}}$, not in a localized/pointwise fashion, but rather in a global way by representing the error based on its inner product against arbitrary $\boldsymbol{v}$. At a specific time $t$, the functional norm of $R(\mathbf{u}_{\mathcal{N}})(t, \cdot) \in \mathcal{V}'$ defined as follows is then an excellent measure of the error to be a candidate for a loss function in view of Theorem 3.1:

$$\|R(\mathbf{u}_{\mathcal{N}})(t, \cdot)\|_{\mathcal{V}'} := \sup_{\boldsymbol{v} \in \mathcal{V}, \|\boldsymbol{v}\|_{\mathcal{V}}=1} |(R(\mathbf{u}_{\mathcal{N}}), \boldsymbol{v})|. \quad (3.3)$$

**Theorem 3.1** (*A posteriori* error bound for any fine-tuned approximations, informal version)**.** *Let the weak solution to (2.2) be $\boldsymbol{u} \in L^2(\mathcal{T}; \mathcal{V})$, and $\partial_t \boldsymbol{u} \in L^2(\mathcal{T}; \mathcal{V}')$. For $\boldsymbol{u}$ that is sufficiently regular, the dual norm of the residual is efficient to estimate the error for any $\mathbf{u}_{\mathcal{N}}$:*

$$\|R(\mathbf{u}_{\mathcal{N}})\|_{L^2(\mathcal{T}; \mathcal{V}')}^2 \lesssim \|\boldsymbol{u} - \mathbf{u}_{\mathcal{N}}\|_{L^2(\mathcal{T}; \mathcal{V})}^2 + \|\partial_t(\boldsymbol{u} - \mathbf{u}_{\mathcal{N}})\|_{L^2(\mathcal{T}; \mathcal{V}')}^2. \quad (3.4)$$

---

[2]Line 205 and 209 in the original FNO3d source code `fourier_3d.py` (link provided for an unaltered fork of the `master` branch commit `de514f2` in the original FNO GitHub repository).

*Moreover, if $\boldsymbol{u}$ and $\boldsymbol{u}_{\mathcal{N}}$ are "sufficiently close", then it is reliable to serve as an error measure:*

$$\|\boldsymbol{u} - \boldsymbol{u}_{\mathcal{N}}\|_{L^{\infty}(\mathcal{T}_m;\mathcal{H})}^2 + \|\boldsymbol{u} - \boldsymbol{u}_{\mathcal{N}}\|_{L^2(\mathcal{T}_m;\mathcal{V})}^2 \leq \left\|(\boldsymbol{u} - \boldsymbol{u}_{\mathcal{N}})(t_m, \cdot)\right\|_{\mathcal{V}}^2 + C \int_{\mathcal{T}_m} \|R(\boldsymbol{u}_{\mathcal{N}})(t, \cdot)\|_{\mathcal{V}'}^2 \, \mathrm{d}t.$$
(3.5)

*where $\mathcal{T}_m := (t_m, t_{m+1}]$, and the constants depend on the regularity of the true solution $\boldsymbol{u}$.*

## 3.2 Fine-tuning using negative Sobolev norm as loss

The functional norm (3.3) is "global" because it does not have a natural $\ell^2$-like summation form where each summand can be evaluated in a localized neighborhood of grid points. Nevertheless, thanks to the Gelfand triple, and viewing the Fourier transform as an automorphism in the tempered distribution space (e.g., see Peetre (1975) and Gel'fand & Shilov (2016, Chapter 3)), we can define the pairing between $\mathcal{V}$ and $\mathcal{V}'$ as follows without getting into the intricate natures of the tempered distribution:

$$\langle f, g \rangle_{\mathcal{V},\mathcal{V}'} \text{ "=" } \int_{\mathbb{Z}^2} (1 + |\boldsymbol{k}|^2)^{-s} \hat{f}(\boldsymbol{k}) \overline{\hat{g}(\boldsymbol{k})} (1 + |\boldsymbol{k}|^2)^s \mathrm{d}\boldsymbol{k}, \text{ where } \hat{v} := \mathcal{F}v \text{ for } v \in \mathcal{V}'.$$
(3.6)

The spatial Sobolev space $H^s(\mathbb{T}^2)$ can be alternatively identified using norm $\|\cdot\|_s$ and seminorm $|\cdot|_s$ (e.g., see Ruzhansky & Turunen (2009, Def. 3.2.2)) as follows for any $s \in \mathbb{R}$

$$\|f\|_s^2 := \sum_{\boldsymbol{k} \in \mathbb{Z}^2} (1 + |\boldsymbol{k}|^2)^s |\hat{f}(\boldsymbol{k})|^2, \text{ and } |f|_s^2 := \sum_{\boldsymbol{k} \in \mathbb{Z}_n^2 \setminus \{\boldsymbol{0}\}} |\boldsymbol{k}|^{2s} |\hat{f}(\boldsymbol{k})|^2, \ s \neq 0.$$
(3.7)

Moreover, we have the subsequent lemma in our specific case.

**Theorem 3.2** (Functional norm "$\simeq$" negative norm). *If $f \in \mathcal{H} := L^2(\mathbb{T}^2)/\mathbb{R}$, $\|f\|_{\mathcal{H}'} = |f|_{-1}$.*

Spatially, we realize a regularized negative Sobolev norm by the Fast Fourier Transform (FFT):

$$\|f\|_{-1,\alpha,n}^2 := \sum_{\boldsymbol{k} \in \mathbb{Z}_n^2 \setminus \{\boldsymbol{0}\}} (\alpha + |\boldsymbol{k}|^2)^{-1} |\hat{f}(\boldsymbol{k})|^2, \text{ where } \mathbb{Z}_n^2 := (\mathbb{Z}/n\mathbb{Z})^2, \text{ and } \alpha \geq 0.$$
(3.8)

With this norm at hand, (3.3) and the Bochner norm in (3.4) of the residual are realized to serve as the loss function in the fine-tuning to "refine" the approximation in the spectral domain

$$\eta_m(\boldsymbol{u}_{\mathcal{N}}, \partial_t \boldsymbol{u}_{\mathcal{N}}) := \|R(\boldsymbol{u}_{\mathcal{N}})(t_m, \cdot)\|_{-1,\alpha,n}.$$
(3.9)

**Parseval identity.** In the context of using an optimization algorithm to train an NN as a function approximator, it is known (e.g., Siegel et al. (2023)) that whether the integral in the loss function is accurately computed affects whether the NN can achieve the scientific computing level of accuracy. For example, on a uniform grid, the accuracy of the mesh-weighted spatial $\ell^2$-norm as the numerical quadrature is highly affected by the *local* smoothness of the function in consideration. Nevertheless, computing the integral in the frequency domain yields exponentially convergent approximations (Trefethen & Weideman, 2014, Theorem 3.1) thanks to the Parseval identity.

**Why functional-type norm for the residual evaluation?** We note that in "physics-informed" learning of operators, for example, in Li et al. (2024d), the PDE residual is evaluated using $L^2$-norm as loss. In the meantime, *positive* Sobolev norm, which is local in terms of differential operators, is used in Li et al. (2022). To our best knowledge, the $H^{-1}$-functional norm has not been applied in either function learning or operator learning problems. In fact, the relation of the Gelfand triple is so simple and elegant, leading to Theorem 3.2, that the functional norm is nothing but an inverse frequency weight in the frequency domain that emphasizes the learning of *low frequency* information. This suits especially well for the learning problem of NSE. Quite contrary to the intuition of FNO variants having the frequency truncation that results error dominating in the high-frequency part, it has been discovered in Lippe et al. (2023) that the error of operator learning for NSE is still dominant in the lower end of the spectrum. This has been corroborated in our study as well, see Figure 8 and Figure 6. For a more detailed and mathematically enriched discussion on why functional norm is not widely popular in traditional numerical methods, we refer the reader to Appendix B.

## 3.3 New training and spectral fine-tuning paradigm

With the trajectory-to-trajectory learner and the error estimators, we propose a new training-fine-tuning paradigm. Another important motivation of this new paradigm is that, experimentally, all

FNO variants capture the statistical properties of the 2D turbulence (Benzi et al., 1987; Boffetta & Ecke, 2012) quickly in training. Even after **a single epoch**, the evaluations of ST-FNO converge to a reasonably small neighborhood of the ground truth in terms of the frequency signature of the energy cascade of the flow (see e.g., Figure 8, and Appendix D.3 for more details). Then, 95% of the FLOPs spent in training only contribute to marginal improvements. These observations motivate us to rethink a more efficient paradigm than end-to-end, in the meantime not needing to initiate the expensive training of another nonlinear corrector as PDE-Refiner (Lippe et al., 2023) in the postprocessing phase.

**The computation of extra fields.**    In evaluating the loss accurately and relying on this evaluation to apply the gradient method, another key barrier is that the extra field variables in evaluating the residual for the velocity (3.2), one needs to compute $\partial_t \boldsymbol{u}_{\mathcal{N}}$. Instead of a common approach of using NO to represent the extra field variables (Wen et al., 2022; Brandstetter et al., 2023), we opt to apply *a traditional implicit-explicit (IMEX) numerical solver $B_\gamma(\cdot)$ with a fine time step* (Wang, 2012), e.g., $\mathcal{O}((\Delta t)^\gamma)$ for $\gamma \geq 2$, to compute these extra field variables (Line 8 of Algorithm 1), while preserving the computational graph for auto-differentiation. We note that, in NSE simulations using traditional numerical solvers, for efficiency, the magnitude of this fine time step is never realistic or attainable due to time marching. Given the training data with trajectories at $\{t_1, \ldots, t_\ell\}$ aiming to predict the trajectories at $\{t_{\ell+1}, \ldots, t_{\ell+n_t}\}$, the new paradigm to train and fine-tune ST-FNO is outlined in Algorithm 1.

---

**Algorithm 1** The new parallel-in-time spectral refining fine-tune strategy in small data regime.

---

**Input:** ST-FNO $G_{\theta,\Theta} := \widetilde{Q}_\theta \circ G_\Theta$; time stepping scheme $B_\gamma(\cdot)$; optimizer $\mathcal{D}(\theta, \nabla_\theta(\cdot))$; training dataset: solution trajectories at $[t_1, \ldots, t_{\ell'}]$ as input and at $[t_{\ell'+1}, \ldots, t_{\ell'+n_t'}]$ as output.

1:  Train the ST-FNO model until the energy signature matches the energy cascade.

2:  Freeze $\Theta$ in $\mathcal{G}_\Theta$ of ST-FNO $\mathcal{G}_{\theta,\Theta}$, keep $\widetilde{Q}_\theta$ trainable.

3:  Cast all `nn.Module` involved and tensors to `torch.float64` and `torch.complex128` hereafter.

**Input:** Evaluation dataset: solution trajectories at $[t_1, \ldots, t_\ell]$ as input, output time step $n_t$.

4:  **for** $m = \ell, \cdots, \ell + n_t - 1$ **do**

5:      Extract the latent fields $\boldsymbol{v}_{\mathcal{N}}^{(m+1)}$ output of $G_\Theta$ at $t_{m+1}$ and hold them fixed.

6:  By construction of ST-FNO: $\boldsymbol{u}_{\mathcal{N}}^{(m+1)}(\theta) := \boldsymbol{u}_{\mathcal{N}}^{(\ell)} + \widetilde{Q}_\theta(\boldsymbol{v}_{\mathcal{N}}^{(m+1)})$ for all $m$.

7:  **for** $j = 1, \cdots, \texttt{Iter}_{\max}$ **do**

8:      March one step with $(\Delta t)^\gamma$ using $B_\gamma$ and approximate the $\partial_t \boldsymbol{u}_{\mathcal{N}}$ as follows:
  $D_t \boldsymbol{u}_{\mathcal{N}}^{(m+1)}(\theta) := (\Delta t)^{-\gamma}(B_\gamma(\boldsymbol{u}_{\mathcal{N}}^{(m+1)}(\theta)) - \boldsymbol{u}_{\mathcal{N}}^{(m+1)}(\theta))$ for all $m$.

9:      Compute $\eta^2 := \sum_m \eta_m^2(\boldsymbol{u}_{\mathcal{N}}^{(m+1)}(\theta), D_t \boldsymbol{u}_{\mathcal{N}}^{(m+1)}(\theta))$ for all evaluation time steps.

10:     Apply the optimizer to update parameters in $\widetilde{Q}$: $\theta \leftarrow \mathcal{D}(\theta, \nabla_\theta(\eta^2))$.

11:     Forward pass only through $\widetilde{Q}$ to update $\boldsymbol{u}_{\mathcal{N}}^{(m+1)} \leftarrow \boldsymbol{u}_{\mathcal{N}}^{(\ell)} + \widetilde{Q}_\theta(\boldsymbol{v}_{\mathcal{N}}^{(m+1)})$ for all $m$.

**Output:** A sequence of velocity profiles at corresponding time steps $\{\boldsymbol{u}_{\mathcal{N}}^{(m)}\}_{m=\ell+1}^{\ell+n_t}$.

---

**Interpretations of the theoretical results.**    Theorem 3.1 establishes that the functional norm of the residual $R(\boldsymbol{u}_{\mathcal{N}})$, without accessing $\boldsymbol{u}$, is a good representation of the error $\boldsymbol{u} - \boldsymbol{u}_{\mathcal{N}}$ in Bochner norms. Estimate (3.4) indicates that reducing the *a posteriori* error estimator is a *necessary* condition for reducing the true error. While estimate (3.5) is more delicate in that it is only reliable when $\boldsymbol{u}_{\mathcal{N}}$ gets "close" to $\boldsymbol{u}$. Theorem 3.2 lays the foundation to accurately evaluate this functional norm via FFT. Nevertheless, (3.5) serves as a guideline to design the "refining" procedure (lines 11 in Algorithm 1), where the error estimator moves to refine the next time step once the error in the previous one becomes less than a given threshold.

## 4    NUMERICAL EXPERIMENTS

### 4.1    ILLUSTRATIVE EXAMPLE: TAYLOR-GREEN VORTEX

In this illustrative example, we examine the 2D Taylor-Green vortex model (Taylor & Green, 1937), whose analytical solution is *known* and frequently employed as a benchmark for evaluating traditional numerical schemes for NSE (Gottlieb & Orszag, 1977). We create a toy train dataset with 10 trajectories on a $256^2$ grid with varying numbers of vortices per wavelength, and the test sample has

an unseen number of vortices yet still can be resolved up to the Nyquist scale. For details please refer to Appendix C.2. The FNO3d used in this example is scaled down to 1 layer.

## 4.2 2D ISOTROPIC TURBULENCE

This meta-example contains a series of examples of isotropic turbulence (McWilliams, 1984), featured in various work such as Kochkov et al. (2021); Li et al. (2021). The energy and the enstrophy spectra satisfy the energy law of turbulence first proposed by A. N. Kolmogorov (Kolmogorov, 1941). The data are generated using a second-order time-stepping scheme that is proven in theory to preserve the inverse cascade of the energy/enstrophy spectra (Wang, 2012; Gottlieb et al., 2012). We consider two cases:

(I) NSE benchmark in Li et al. (2021), $\nu = 10^{-3}$, $\omega_0 \sim \mathcal{N}(0, (-\Delta + \tau^2 I)^{-\alpha/2})$, the energy density in wavenumber $k := |\boldsymbol{k}|$ is $f(k) \sim (k^2 + \tau^2)^{-\alpha}$, no drag, fixed force with a low wavenumber.

(II) The famous decaying turbulence discovered in McWilliams (1984), the initial power spectrum $|\hat{\psi}(k)|^2 \sim k^{-1}(\tau^2 + (k/k_0)^4)^{-1}$ is chosen that the decaying is slow and the enstrophy density evolves to a similar energy cascade to the Kolmogorov flow used as examples in Kochkov et al. (2021); Lippe et al. (2023); Sun et al. (2023).

Tables 2 and 3 report the results for example (I) and example (II), respectively. In Appendix D.2,Table 5 and Table 6 report the architectural changes and the computational efficiency comparison with roll-out and traditional solver.

## 5 CONCLUSION AND LIMITATIONS

We designed a new pipeline to train and fine-tune a new spatiotemporal modification of FNO to get close to the true solution (not the ground truth generated by the numerical solver) of NSE in the turbulent regime. The evaluation errors in benchmark problems are up to $10^5$ times better than the non-fine-tuned FNO variants trained by a simple end-to-end pipeline. Due the exploitation of the intricate connections with traditional spectral methods, e.g., the optimally learned parameters correspond to a Fourier-Galerkin projection (Kovachki et al., 2021) using the Fourier basis, only FNO-based neural operators benefit from this advancement. How to generalize this new pipeline to other types of operator learners in Appendix B will be worthy of exploration.

Table 1: Results for Taylor-Green vortex, the relative errorsª at the final time step, $\varepsilon := \omega_{\text{true}} - \omega_{\mathcal{N}}$.

|  | Evaluation after training | | After fine-tuning | |
| --- | --- | --- | --- | --- |
|  | $\|\varepsilon\|_{L^2}$ | $\|R\|_{-1,n}$ | $\|\varepsilon\|_{L^2}$ | $\|R\|_{-1,n}$ |
| ST-FNO3d | $1.94\times10^{-1}$ | $2.18\times10^{-1}$ | $1.24\times10^{-7}$ | $3.21\times10^{-7}$ |
| PS+RK2 (GT) | $5.91\times10^{-6}$ | $1.16\times10^{-5}$ | N/A | N/A |

Table 2: Ablation study using forced turbulence, original example from Li et al. (2021). $\varepsilon := \omega_{\mathcal{S}} - \omega_{\mathcal{N}}$

| ST-FNO3d | Evaluation after training | | After fine-tuning | |
| --- | --- | --- | --- | --- |
|  | $\|\varepsilon\|_{L^2}$ | $\|R\|_{-1,n}$ | $\|\varepsilon\|_{L^2}$ | $\|R\|_{-1,n}$ |
| 10 ep & $L^2$ FT | $2.08\times10^{-2}$ | $1.27\times10^{-2}$ | $2.82\times10^{-4}$ | $2.78\times10^{-5}$ |
| 10 ep & $H^{-1}$ FT | – | – | $3.16\times10^{-4}$ | $4.59\times10^{-7}$ |

Table 3: Evaluation metrics of the McWilliams isotropic turbulence example. All models are trained using on $64\times64$ mesh and evaluated on $256\times256$ mesh. $\varepsilon := \boldsymbol{u}_{\mathcal{S}} - \boldsymbol{u}_{\mathcal{N}}$ or $\omega_{\mathcal{S}} - \omega_{\mathcal{N}}$. $\mathcal{S}$ stands for the Fourier approximation space (3.1) on a $256\times256$ fine grid. $r := R(\boldsymbol{u}_{\mathcal{N}})$ or $R(\omega_{\mathcal{N}})$. $\mathcal{Y} := H^{-1}(\mathbb{T}^2)$. All error norms are evaluated at the final time step. $H^{-1}$ appending model means the training uses the difference in the $H^{-1}$-norm as the loss function. Errors are measured for 32 trajectories in the test dataset. Fine-tuning uses 100 iterations of Adam optimizer in line 10.

|  | Evaluation after training | | After fine-tuning | |
| --- | --- | --- | --- | --- |
|  | $\|\varepsilon\|_{L^2}$ | $\|R\|_{-1,n}$ | $\|\varepsilon\|_{L^2}$ | $\|R\|_{-1,n}$ |
| FNO3d 10 ep, $L^2$ train, no FT | $4.32\times10^{-2}$ | $6.09\times10^{-2}$ | N/A | N/A |
| FNO3d 100 ep, $L^2$ train, no FT | $3.87\times10^{-2}$ | $2.44\times10^{-2}$ | N/A | N/A |
| ST-FNO3d 10 ep, $H^{-1}$ train & $H^{-1}$ FT | $6.54\times10^{-2}$ | $6.19\times10^{-2}$ | $5.71\times10^{-3}$ | $2.24\times10^{-5}$ |
| ST-FNO3d 10 ep, $L^2$ train & FT | $3.69\times10^{-2}$ | $2.35\times10^{-2}$ | $1.79\times10^{-3}$ | $9.55\times10^{-5}$ |
| ST-FNO3d 10 ep, $L^2$ train & $H^{-1}$ FT | – | – | $2.88\times10^{-4}$ | $4.02\times10^{-6}$ |

## ACKNOWLEDGMENTS

The research of Brarda and Xi is supported by the National Science Foundation award DMS-2208412. The work of Li was performed under the auspices of the U.S. Department of Energy by Lawrence Livermore National Laboratory under Contract DEAC52-07NA27344 (LLNL-CONF-872321) and was supported by the LLNL-LDRD program under Project No. 24ERD033. Cao is supported in part by the National Science Foundation award DMS-2309778.

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

## A  NOTATIONS

Table 4: Notations used in an approximate chronological order and their meaning in this work.

| Notation | Meaning |
|---|---|
| $\Omega$ | $\Omega = (0, 2\pi)^2$ or $(0,1)^2$ the modeling domain in $\mathbb{R}^2$ |
| $\mathcal{V}, \mathcal{H}$ | Hilbert spaces defined on the domain $\Omega$ above, $f \in \mathcal{H} : \Omega \to \mathbb{R}$ |
| $\mathcal{X}, \mathcal{Y}$ | product spaces $\prod_{j \in \Lambda} \mathcal{H}$ |
| $\|$ | concatenation, for $f_j \in \mathcal{H}, \|_{j \in \Lambda} f_j \in \prod_{j \in \Lambda} \mathcal{H}$ |
| $\nu$ | viscosity, the inverse of the Reynolds number, strength of diffusion |
| $\mathbb{T}^2$ | $\mathbb{T}^2 := S^1 \times S^1$ where $S^1$ is homeomorphic to $[0,1)$ with the point 1 being 0. |
| $H^1(\mathbb{T}^2)$ | $\{u \in H^1(\Omega) : u \text{ satisfies the periodical boundary condition}\}$ |
| $\boldsymbol{H}^1(\mathbb{T}^2)$ | $\boldsymbol{H}^1(\mathbb{T}^2) := H^1(\mathbb{T}^2) \times H^1(\mathbb{T}^2)$ |
| $\mathbf{rot}\,\psi$ | $\mathbf{rot}\,\psi := (\partial_y \psi, -\partial_x \psi)$ is the rotated gradient vector |
| $L^p(\mathcal{T}; \mathcal{V})$ | the Bochner spaces containing $\{u : \mathcal{T} \to \mathcal{V} \mid \int_{\mathcal{T}} \|u(t, \cdot)\|_{\mathcal{V}}^p \, \mathrm{d}t < +\infty\}, p \in [1, \infty)$ |
| $L^\infty(\mathcal{T}; \mathcal{V})$ | Bochner space containing $\{u : \mathcal{T} \to \mathcal{H} \mid \mathrm{ess\,sup}_{t \in \mathcal{T}} \|u(t, \cdot)\|_{\mathcal{V}} < +\infty\}$ |
| $\boldsymbol{u}$ | the true weak solution to the NSE $\boldsymbol{u}(t, \cdot) \in \mathcal{V}$ for any $t$ |
| $\boldsymbol{u}_{\mathcal{S}}^{(l)}$ | the ground truth data generated at $t_l$ by the numerical solver in $\mathcal{S} \subset \mathcal{V}$ |
| $\boldsymbol{u}_{\mathcal{N}}^{(m)}$ | neural operator evaluations at $t_m$ that can be embedded in $\mathcal{S} \subset \mathcal{V}$ |
| $(u, v)$ or $(\boldsymbol{u}, \boldsymbol{v})$ | the $L^2$-inner product on $\mathcal{H}$, $(u, v) := \int_\Omega uv \, \mathrm{d}\boldsymbol{x}$ |
| $\|u\|_s$ | the $H^s$-norm of $u$, computed by (3.7), $\|u\| := \|u\|_0$ falls back to $L^2$-norm |
| $|u|_s$ | the $H^s$-seminorm of $u$, computed by (3.7), is a norm on $H^s(\mathbb{T}^2)/\mathbb{R}$ |
| $\lesssim$ | $a \lesssim b$ means that $\exists c$ independent of asymptotics if any such that $a \leq cb$ |
| $\simeq$ | $a \simeq b \Leftrightarrow a \lesssim b$ and $b \lesssim a$ |
| $\mathcal{V}'$ | dual space of $\mathcal{V}$, contains all continuous functionals $f$ such that $|f(v)| \lesssim \|v\|_{\mathcal{V}}$ |
| $\langle f, v \rangle$ | the pairing between $v \in \mathcal{V}$ and $f \in \mathcal{V}'$, can be identified as $(f, v)$ if $f \in \mathcal{H}$ |
| $\mathcal{V} \hookrightarrow \mathcal{H}$ | $\mathcal{V}$ is continuously embedded in $\mathcal{H}$ such that $\|u\|_{\mathcal{H}} \lesssim \|u\|_{\mathcal{V}}$ |
| $\mathcal{V} \Subset \mathcal{H} \Subset \mathcal{V}'$ | compact embeddings by Poincaré inequality, $\mathcal{H} = L^2(\mathbb{T}^2), \mathcal{V} = H^1(\mathbb{T}^2)$ |
| $\boldsymbol{A} : \boldsymbol{B}$ | $\boldsymbol{A} : \boldsymbol{B} = \sum_{1 \leq i,j \leq 2} A_{ij} B_{ij}$ for $\boldsymbol{A}, \boldsymbol{B} \in \mathbb{R}^{2 \times 2}$ |
| $\boldsymbol{a} \otimes \boldsymbol{b}$ | $\boldsymbol{a}\boldsymbol{b}^\top$ if both are viewed as column vectors |

## B  DETAILED LITERATURE REVIEW AND MOTIVATIONS

**Interplay of deep learning and PDEs.**    PDE solvers are function learners to represent PDE solutions using neural networks (Han et al., 2018; Raissi et al., 2019; Chen & Koohy, 2024). PDE discovery encompasses all the techniques dedicated to identifying and optimizing PDE coefficients from data (Brunton et al., 2016; Champion et al., 2019). Recently, hybrid solver approach has been explored in Greenfeld et al. (2019); Kochkov et al. (2021); Huang et al. (2023); Taghibakhshi et al. (2023; 2021); Hu & Jin (2024); Huang et al. (2022). Reinforcement learning has been applied in the field of mesh generation to build more efficient solving pipelines (Yang et al., 2023; Gillette et al., 2024). For neural operators, remarkable outcomes are achieved in diverse applications, such as weather forecasting (Keisler, 2022) and turbulent fluids simulations (Shu et al., 2023; Li et al., 2023a; Lienen et al., 2023; Li et al., 2025), the methods based on neural operators have significantly influenced the progress of the interplay between PDE and deep learning. This success was a natural outcome of several improvements brought to the field, for example, fast solution evaluations, a feature very appealing in many engineering applications (Zheng et al., 2023); and the ability to provide mesh-free and resolution-independent solvers in cases of irregular domains (Hao et al., 2023; Li et al., 2024c).

**Neural operators.**    Looking at neural operator architectures, we can identify two different approaches. The first one creates "basis" (or frames) through nonlinear approximators and aggregates them linearly. Developed architectures that belong to this category include Deep Operator Network (DeepONet) (Lu et al., 2021; Goswami et al., 2023; Wang et al., 2021), wherein aggregation occurs via linear combination. Similarly, Fourier Neural Operator (FNO) (Li et al., 2021), and some of its variants (Gupta et al., 2021; Rahman et al., 2023; Tran et al., 2023; Li et al., 2024c; Wen et al., 2022), aggregate different latent representations through convolution with a learnable kernel in the

frequency domain. Additional examples with a linear aggregation with a U-Net meta-architecture can be found in Raonic et al. (2024); He et al. (2024). Bartolucci et al. (2024) further explores the error analyses for the linear aggregations through the lens of the frame(let) theory. The second approach to designing neural operator architectures aims to obtain the "basis" through linear projections of the latent representations. In this scenario, the aggregation is nonlinear, for example, using a signal-dependent kernel integral. In this category, the most notable example is the scaled dot-product attention operator in Transformer (Vaswani et al., 2017), which builds an instance-dependent kernel. In the context of operator learning applications, Transformer-based neural operators have been studied in Cao (2021); Kissas et al. (2022); Li et al. (2023b); Hao et al. (2023); Wu et al. (2023); Li et al. (2024a); Fonseca et al. (2023); Li et al. (2024b); Calvello et al. (2024); Liu et al. (2024); Xiao et al. (2024); Hao et al. (2024); Wu et al. (2024a); Shih et al. (2025). It is also shown in Lanthaler et al. (2023) that nonlinear aggregations outperform its linear counterparts in learning solutions with less regularity. More recently, for spatiotemporal problems, the state-space based neural operators explore an expansion to the natural operator exponential representation of the solution map, see e.g., Zheng et al. (2024); Cheng et al. (2024); Hu et al. (2024); Ruiz et al. (2024).

**Numerical methods for NSE.** The NSE falls into the category of a stiff PDE (system) $\partial_t u = Lu + N(u) + f$, where $f$ is the external forcing, $L$ and $N(\cdot)$ are a linear and a nonlinear operator, respectively. Trefethen noted back in Kassam & Trefethen (2005) on the difficulty to design a time-stepping scheme as $N(\cdot)$ and $L$ have to be treated differently. There are a long history of numerical methods for NSE we draw inspiration from. Petrov-Galerkin methods have been developed for NSE in Boffi et al. (2013); Girault & Raviart (2012). Nonlinear Galerkin method (Marion & Temam, 1990) inspires us to prove Theorem E.9. Chorin (1968) uses a clever trick to impose the divergence free condition without constructing a divergence-conforming finite element subspace. Shen (1994) designed various Galerkin methods in the space of orthogonal polynomials. Pioneered by Chorin, Shen, and E, projection methods are among the most popular schemes to solve NSE (see e.g., Weinan & Liu (1995) for a summary), also serves as an inspiration to add the Helmholtz layer. Bernardi et al. (1992) proposed a mixed discretization for the vorticity-streamfunction formulation.

**Consistency-stability trade-offs.** In view of the Lax equivalence principle ("consistency" + "stability" $\implies$ "convergence" (Lax, 2002, Theorem 8)), the improvement of the stability of the method has a trade-off with a method's consistency at the cost of the approximation capacity. Numerical methods for NSE is the epitome for such trade-off. For example, high-order explicit time stepping schemes offer better local truncation error estimates near boundary layers of the flow (Lele, 1992), yet the lack of stability is more severe and needs higher-order temporal smoothing. On the other hand, implicit schemes can be unconditionally stable for stiff or highly transient NSE with relatively large time steps $\mathcal{O}(1)$. The stability in implicit schemes becomes much less stringent on the time step, as no CFL condition is required. However, the solution at the next time step requires solving a linear or nonlinear system. Thus, computationally implicit schemes are usually an order of magnitude more expensive compared to explicit schemes. The implicit-explicit (IMEX) stable solver chosen in the fine-tuning postprocessing is from Wang & Liu (2002); Wang (2012).

**De-aliasing filter sacrifices consistency for stability.** One famous example of the consistency-stability trade-off is the $3/2$-rule (also known as $2/3$-dealiasing filter) for the nonlinear convective term (Orszag, 1971b; Patterson & Orszag, 1971; Hou, 2009; Gottlieb & Orszag, 1977) for pseudo-spectral methods (Orszag, 1971a; 1972). The highest $1/3$ modes, the inclusion of which contributes to better approximation capacity, are filtered out to ensure long-term stability. Compromises such as the CFL condition and the $3/2$-rule *must* be made for traditional numerical schemes to be stable, keeping the balance between stability and accuracy. These constraints apply to traditional numerical methods because any solver has to march a consecutive multitude of time steps, which makes the error propagation operator's norm a product of many. Numerical results suggest that for pseudo-spectral spatial discretization with no higher-order Fourier smoothing temporally, the dealiasing filter is indispensable (Tadmor, 1987), as the time marching may experience numerical instability (Kreiss & Oliger, 1979; Goodman et al., 1994) without it. This is due to the nonlinear interaction in the convective term, which is caused by the amplification of high-frequency "aliasing" errors when the underlying solution lacks sufficient smoothness. In this study, the hybrid approach we adopted combines the strengths of NOs and traditional numerical solvers. There is no time marching

consecutively for a multitude of time steps, which renders the method free of the stability constraints such as the CFL condition and the $3/2$-rule.

**Why and why not functional-type a posteriori error estimation?** For the sake of presentation, we make some handy assumptions: (1) $A : \mathcal{V} \to \mathcal{V}'$ denotes a linear operator on the Gelfand triple; (2) $u_\mathcal{S}$ is obtained through Galerkin methods (Evans, 2022), which is a projection onto a finite-dimensional approximation subspace of $\mathcal{S} \subset \mathcal{V}$; and (3) an evaluation of $u_\mathcal{N}$ can be continuously embedded into $\mathcal{S}$. Then, we have the following enlightening Pythagorean-type identity for the true solution $u$ satisfying $Au = f$

$$\|u - u_\mathcal{S}\|_a^2 + \|u_\mathcal{S} - u_\mathcal{N}\|_a^2 = \|u - u_\mathcal{N}\|_a^2 \quad \text{since } (u - u_\mathcal{S}) \perp_a (u_\mathcal{S} - u_\mathcal{N}) \in \mathcal{V}. \tag{B.1}$$

Through the Riesz representation theorem, $u_\mathcal{S}$ is solved through $(u - u_\mathcal{S}, v)_a := \langle A(u) - A(u_\mathcal{S}), v \rangle = 0$, for any $v \in \mathcal{S}$. This bilinear form $(\cdot, \cdot)_a$ induces a (semi)norm $\| \cdot \|_a$ inheriting the topology from $\mathcal{V}$. Given this orthogonality, minimizing the difference between $u_\mathcal{N}$ and $u_\mathcal{S}$ becomes fruitless if $\|u - u_\mathcal{S}\|_a$ is relatively big in the first place, and unnecessary computational resources may have been spent to get closer to $u_\mathcal{S}$. Rather, (B.1) indicates that it is more efficient if one can design a method to reduce the error of $\|u - u_\mathcal{N}\|_a$ directly, while circumventing the fact that $u$ is not accessible.

Speaking of the *a posteriori* error estimation in traditional PDE discretization, part of the goal is to help the adaptive mesh refinement to get a better local basis. In this regard, the global error functional in negative Sobolev spaces must be approximated using localized $L^2$ residuals to indicate where the mesh needs to be refined. There are various compromises for this $H^{-1}$-to-$L^2$ representation to happen that renders the estimate inaccurate, such as discrete Poincaré constant (Veeser & Verfürth, 2012) or inverse inequalities (Carstensen & Funken, 1999; Veeser & Verfürth, 2009), see also Verfürth (2013, §1.6.2). Functional-type *a posteriori* error estimation (Repin, 2008) consider the error as a functional, which is equivalent to error to bilinear form-associated norm as follows

$$R(u_\mathcal{S}) \in \mathcal{V}', \text{ and } \langle R(u_\mathcal{S}), v \rangle = (u - u_\mathcal{S}, v)_a,$$

where the weak form of the PDE is

$$(u, v)_a = f(v) \,\forall v \in \mathcal{V} \text{ and } (u_\mathcal{S}, v)_a = f(v) \,\forall v \in \mathcal{S} \subset \mathcal{V}.$$

The common approach is using the help from extra "flux" or "stress" dual variables (Repin, 2008, §6.4), see also Repin (2000) for how to get an accurate representation of the error functional in $L^2$. For example, for the Stokes flow, which can be viewed as a steady-state limit of viscous fluid, such that $\partial_t \boldsymbol{u} = 0$ and no nonlinear convection in (2.2), $Re \sim \mathcal{O}(1)$, Repin (2008, §6.2) estimates error of an $\boldsymbol{H}(\text{div})$–$L^2$ mixed discretization as follows

$$\nu\|\nabla(\boldsymbol{u} - \boldsymbol{u}_\mathcal{S})\| \leq \|\boldsymbol{\sigma} + q\boldsymbol{I} - \nu\nabla\boldsymbol{u}_\mathcal{S}\| + C_P\|\nabla \cdot \boldsymbol{\sigma} + \boldsymbol{f}\|,$$

where $C_P$ is the Poincaré constant of the compact embedding and $(\boldsymbol{\sigma}, q)$ is a reconstruction field pair. However, the drawback of this approach is that an expensive global minimization problem needs to be solved, e.g., for $(\boldsymbol{\sigma}, q)$ above. Another main reason to introduce extra field variables is that, for finite element methods, the basis functions are local and do not have a globally continuous derivative, in that $\Delta\boldsymbol{u}_\mathcal{S}$ yields singular distributions, whose proper norm is $H^{-1}$-norm and cannot be evaluated by summing up element-wise $L^2$-norms. In computation, it has to be replace by $\nabla \cdot \boldsymbol{\sigma}$ where $\boldsymbol{\sigma} \in \boldsymbol{H}(\text{div})$, and is constructed to be closer to the true solution's gradient $\nabla\boldsymbol{u}$ than $\nabla\boldsymbol{u}_\mathcal{S}$. Meanwhile, for systems like NSE, the error estimation in Galerkin methods for NSE is further complicated by its saddle point nature from the divergence-free constraint, in that consistency has to be tweaked to ensure the Ladyzhenskaya-Babuška-Brezzi stability condition (Girault & Raviart, 2012, Chapter III §1 Sec 1.2). Recently, Fanaskov et al. (2024) considers NN as a function learner to represent solutions of linear convection-diffusion equations, yet still falls into the traditional error estimation framework in that an extra flux variable is learned by NN, and the error if of typical accuracy of NN function learners $\mathcal{O}(10^{-3})$. In contrast, in our study, the need of extra "stress" or "flux" variables to build the residual functional is circumvented as well.

**Error correction for Bayesian inverse problems.** The *a posteriori* error-correction approaches through sampling for the output of surrogate NNs in Bayesian inverse problems (Yan & Zhou, 2020). More recently, in Cao et al. (2023), the error equation in an inverse problem is approximated by solving for a Galerkin projection of a linearized error equation, while in this paper, we leverage the Gelfand triple directly to compute a nonlinear Galerkin projection directly by minimizing the spectral norm.

**Sobolev norms in operator learning**     In the context of function learning problems, Du et al. (2024) applied an $L^2$-spectral loss for Physics-informed Neural Networks (PINN) (Raissi et al., 2019) and found superior results over mesh-weighted spatial losses. Du et al. (2024) realized the PDE residual as a functional on $L^2$ and exploited the Parseval identity on this space, which computes $\sup_{v \in L^2} \langle R, v \rangle / \|v\|_{L^2}$. We note that this is **not** the natural space to measure the PDE residual according to the Galerkin formulation of NSE. Using the Poisson equation $-\Delta u = f$ as a simpler example, if $u \in H^1$, then the residual $f + \Delta u \in H^{-1}$. $-\Delta$ is a bounded operator from $H^1$ to $H^{-1}$ which guarantees the stability estimate $\|u\|_1 \leq c\|f\|_{-1}$, while $-\Delta$ is **unbounded** from $L^2$ to $L^2$. The dual space of $H^1(\mathbb{T}^2)$ is the natural space to consider the residual functional. The reason is that the solution to NSE belongs to $H^1(\mathbb{T}^2)$ at a given snapshot. The dual space of $H^1(\mathbb{T}^2)$ happens to be $H^{-1}$ using the Gelfand triple. In this paper, we compute $\sup_{v \in H^1} \langle R, v \rangle / \|v\|_{H^1}$. However, this $H^{-1}$-norm is a non-localizable functional norm in the spatial domain, and traditional wisdom (FEM or FVM) has to circumvent the direct evaluation of it by localizing the residual, which introduced inaccurate compromises in the residual-based error estimation.

**A posteriori error estimation for flow problems**     There is a vast amount of literature for the *a posteriori* error estimation for the viscous flow ($1/\text{Re} > 0$) problems under the Hilbertian framework for traditional numerical methods. Our methods draw inspiration from these pioneers and try to address the drawbacks. The most popular *a posteriori* error estimator for stationary Stokes problem is from Verfürth (1989), and it is of residual-type by computing a mesh-weighted $L^2$-norm elementwise, and the singular distribution $\Delta u_S$ is represented by the magnitude of flux jumps across the facets in this mesh. In Bank & Welfert (1991), a more accurate *a posteriori* estimation technique is invented for Stokes problem in which a local problem is solved to represent the residual functional on a collection of neighboring elements. $L^2$ residual-type error estimation for stationary NSE is considered in Oden et al. (1994). To our best knowledge, no functional-type *a posteriori* error estimation has been applied to solve the transient NSE, due to its in-efficiency for traditional finite element or finite volume methods. As at every time step, a global nonlinear problem has to be solved if one ought to evaluate the functional accurately using finite element local basis functions, whose computational cost is an order of magnitude higher than implicit Euler methods.

**Hybrid methods for turbulent flow predictions.**     There are quite a few approaches that combine NN-based learners with traditional time-marching solvers. Notable examples include: solution from Direct Numerical Simulation (DNS) corrected using an NN (Um et al., 2020); interpolation to achieve spatial super-resolution using solutions produced by traditional numerical solvers marching on coarser grids (Kochkov et al., 2021). Sun et al. (2023) uses orthogonal polynomial features (HiPPO from Gu et al. (2020)) to achieve temporal super-resolution in addition to the spatial one. McGreivy & Hakim (2023) explores how to incorporate energy-preserving schemes such as upwinding into the ML solvers through traditional numerical methods. Dresdner et al. (2023) explores the possibility of using neural networks to correct the result of the traditional time-marching solvers. Qi & Sun (2024) identifies important frequency bands in FNO parametrization and improves FNO's rolling-out performance in NSE benchmarks. Wu et al. (2024b) uses graph-based neural ODE to improve the out-of-distribution performance in fluid dynamics tasks.

## C   DATA GENERATION

### C.1   VORTICITY–STREAMFUNCTION FORMULATION

Here we derive the vorticity-streamfunction formulation for the convenience of the readers. Consider the standard NSE in 3D, $\boldsymbol{u}$'s $z$-component is 0, and $\boldsymbol{u} = \boldsymbol{u}(t, x, y)$ has only planar dependence

$$\begin{cases} \dfrac{\partial \boldsymbol{u}}{\partial t} + (\boldsymbol{u} \cdot \nabla)\boldsymbol{u} = -\dfrac{1}{\rho}\nabla p + \nu \Delta \boldsymbol{u} + \boldsymbol{f}. \\ \nabla \cdot \boldsymbol{u} = 0 \end{cases} \qquad (C.1)$$

Taking $\nabla \times (\cdot)$ on both sides, and we assume that the solution is sufficiently regular that the spatial and temporal derivative can interchange, as well as curl and the Laplacian can interchange, one gets the following equation by letting $\boldsymbol{\omega} = \nabla \times \boldsymbol{u}$

$$\frac{\partial \boldsymbol{\omega}}{\partial t} + \nabla \times \nabla \left(\frac{\boldsymbol{u}^2}{2}\right) + \boldsymbol{u} \cdot \nabla \boldsymbol{\omega} - \boldsymbol{\omega} \cdot \nabla \boldsymbol{u} = -\nabla \times \left(\frac{1}{\rho}\nabla p\right) + \nu \Delta \boldsymbol{\omega} + \nabla \times \boldsymbol{f}. \qquad (C.2)$$

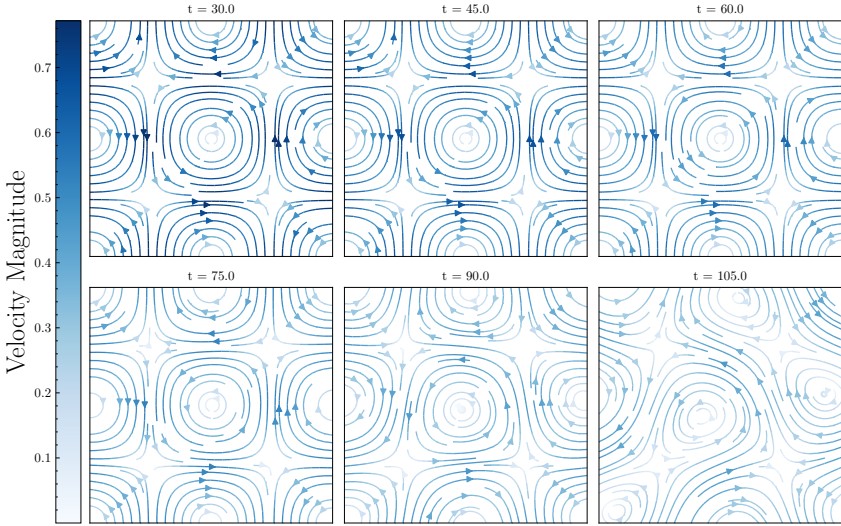

Figure 4: Ground truth streamlines for Taylor-Green vortex example.

Next, one can have $\boldsymbol{\omega} \cdot \nabla \boldsymbol{u} = 0$, thus the equation becomes

$$\frac{\partial \boldsymbol{\omega}}{\partial t} + \boldsymbol{u} \cdot \nabla \boldsymbol{\omega} = \boldsymbol{\omega} \cdot \nabla \boldsymbol{u} + \nu \Delta \boldsymbol{\omega} + \nabla \times \boldsymbol{f}. \tag{C.3}$$

We need to introduce a streamfunction $\psi$ to recover the velocity $\boldsymbol{u} = \nabla \times (0, 0, \psi)$. The main simplification comes from the assumption of dependence on $x$- and $y$-variable only. Thus, in 2D this becomes $\boldsymbol{u} = \mathbf{rot}\,\psi$ (a $\pi/2$ rotation of the 2D gradient $\nabla \psi$). Therefore,

$$\boldsymbol{u} = (u_1, u_2, 0) \implies u_1 = \partial_y \psi, u_2 = -\partial_x \psi$$

and the vector vorticity can be equivalently represented by a scalar vorticity $\omega$

$$\boldsymbol{\omega} = \nabla \times \boldsymbol{u} = (0, 0, \mathrm{curl}(u_1, u_2)) = (0, 0, \partial_x u_2 - \partial_y u_1) =: (0, 0, \omega)$$

The equation becomes a nonlinear system for $\omega$ and $\psi$:

$$\begin{cases} \partial_t \omega + \mathbf{rot}\,\psi \cdot \nabla \omega - \nu \Delta \omega = \nabla \times \boldsymbol{f}, & \text{(C.4)} \\ \omega + \Delta \psi = 0. & \text{(C.5)} \end{cases}$$

## C.2 TAYLOR-GREEN VORTEX

Taylor-Green vortex serves as one of the most well-known benchmarks for NSE numerical methods, as its flow type experiences from laminar, to transitional, and finally evolving into turbulent regime. We consider the trajectory before the breakdown phase. We opt to use a doubly periodic solution on $[0, 2\pi]^2$ such that the inflow/outflow occurs on the "boundary" (see Figure 4). The exact solution is given by:

$$\boldsymbol{u}(t, x, y) = e^{-2\kappa^2 \nu t} \begin{pmatrix} \sin(\kappa x) \cos(\kappa y) \\ -\cos(\kappa x) \sin(\kappa y) \end{pmatrix} \text{ or } e^{-2\kappa^2 \nu t} \begin{pmatrix} -\cos(\kappa x) \sin(\kappa y) \\ \sin(\kappa x) \cos(\kappa y), \end{pmatrix}$$

where $\nu = 1/\mathrm{Re}$ is chosen as $10^{-3}$. For a sample trajectory with $\kappa = 1$, please refer to Figure 4. We apply the pseudo-spectral method with RK2 time stepping for the convection term and Crank-Nicolson for the diffusion term. The dataset has 11 trajectories with $\kappa$ ranging from 1 to 11, among which $\kappa = 11$ is chosen as the test trajectory.

## C.3 2D ISOTROPIC TURBULENCE

The example featured in the original FNO paper (Li et al., 2021) can be viewed as a special case of the isotropic turbulence with a strongly regularized initial condition, no drag, and a force with low

wavenumber. This is our example (I). The example (II) has weakly-regularized initial condition, small drag, and no external force. The initial condition spectra are chosen such that, after the warmup time, the spectra are the comparable with the Kolmogorov flow featured in Kochkov et al. (2021). The similarity is in the sense of the energy cascade formed in the Fourier domain (see Figure 8 and 9). The training data used in both (I) and (II) are generated using a pseudo-spectral method with a de-aliasing filter using an RK4 marching for the convection term and implicit Euler for the diffusion term. The Reynolds number used in this example is 1000. The time steps for both are $\Delta t = 10^{-3}$ on a $256^2$ grid, then downsampled to a $64^2$ grid. The warmup time is 4.5. The $\delta t$ for the ST-FNO training and evaluations are $\delta t = 55\Delta t$. The testing data are generated on a $512^2$ grid with a $5 \times 10^{-4}$ time step then downsampled to $256^2$. There are 1024 trajectories in the training data. The input of model has 10 snapshots from $t = 4.5$ to $t = 4.5 + 9\,\delta t$. The output ground truth has the 10 subsequent snapshots from $t = 4.5 + 10\,\delta t$ to $t = 4.5 + 19\,\delta t$. There are 32 trajectories in the testing data. Each of the testing data's trajectory runs from the same time interval, and has 40 snapshots in the input and 40 snapshots in the output. The initial energy distribution for the training and testing data trajectories are draw from the same energy distribution, see (D.1). For additional experiments using out-of-distribution data with different initial energy distribution and evaluation using trajectories with different Reynolds number than the training data, we refer the reader to Section D.4.

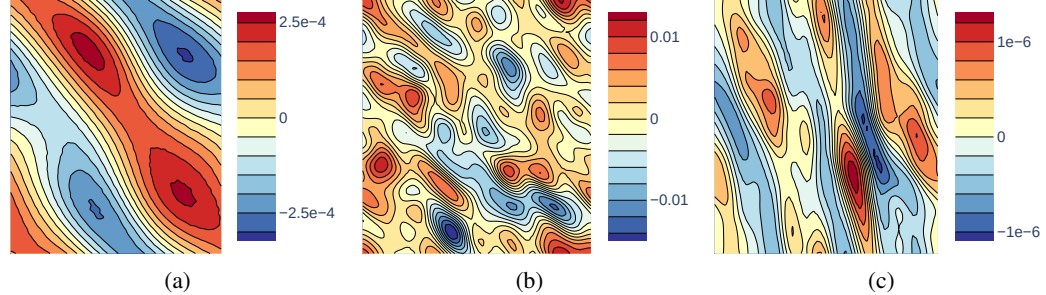

Figure 5: Contours plots of pointwise values of residuals for Example (I). (a): the residual of the ground truth; (b): residual of ST-FNO, where the time derivative in the residual is using the ground truth's; (c): the residual after fine-tuning for 10 ADAM iterations where the time derivative is computed using an extra-fine-step numerical solver.

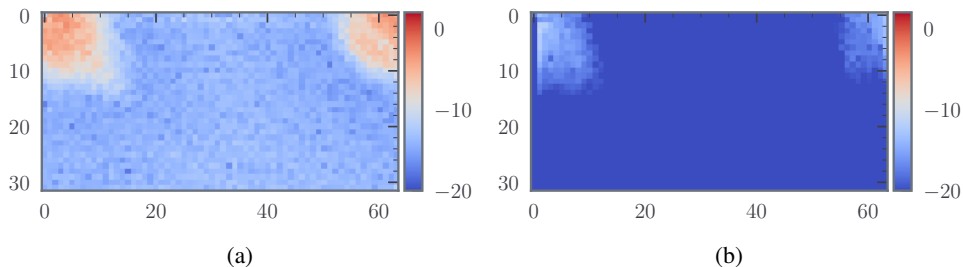

Figure 6: Pointwise values of residual of ST-FNO in the frequency domain in $\ln$ scale of Example (I). (a): the residual of ST-FNO before correction, time derivative computed using the ground truth; (b): the residual of ST-FNO after fine-tuning for 10 ADAM iterations where the time derivative is computed using an extra-fine-step numerical solver.

# D EXPERIMENTS

## D.1 TRAINING AND EVALUATION

The training uses `torch.optim.OneCycleLR` (Smith & Topin, 2019) learning rate strategy with a 20% warm-up phase. AdamW is the optimizer with no extra regularization. The learning rate starts and ends with $10^{-3} \cdot lr_{\max}$. The $lr_{\max} = 10^{-3}$. Despite using the negative Sobolev norm is quite

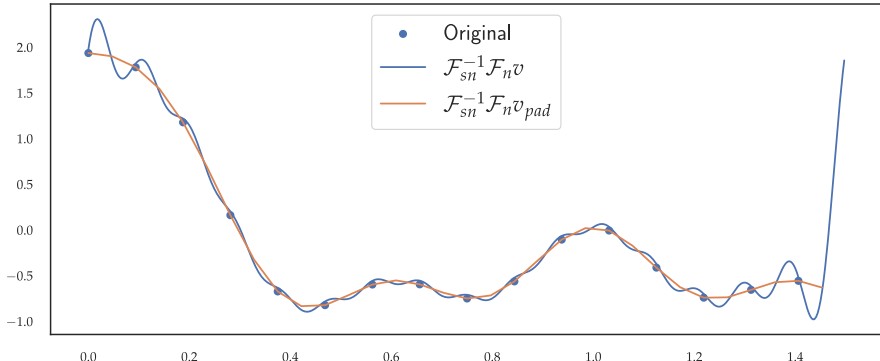

Figure 7: A simple illustration of necessity of padding using a 1D non-periodic data. FNO3d cannot handle temporal non-periodic data. The free temporal super-resolution by FFT/iFFT has Runge-like phenomena near the end points without padding. Padding in the ST-FNO3d modification resolves this problem. Since data themselves are intrinsically lower-dimensional, we opt not to pad the latent representations, but rather apply the padding only in the last spectral convolution layer to get the output.

efficient in fine-tuning, using it in training not be the most efficient in minimizing the $L^2$-norm due to the optimization being non-convex, we observe that all norms are equivalent "bad" due to nonlinearity of NOs. The result demonstrated is obtained from fixing the random number generator seed to be 1127825. All models are trained on a single RTX A4500 or RTX A6000, the inference is done with a power limiter on the GPU to ensure fairness. The codes to replicate the experiments are open-source and publicly available at https://github.com/scaomath/torch-cfd.

### D.2 MODELS

**Helmholtz layer for the velocity-pressure formulation.** For the V-P formulation in a simplified connected convex or periodic domain, it is known that an exact divergence-free subspace $\mathcal{W} \subset \mathcal{V}$ for velocity means that the pressure field is not needed. The reason is that in the weak formulation, the pressure is a Lagrange multiplier to impose the divergence-free condition (Girault & Raviart, 2012, Chapter III 1 Section 1.1). Inspired by the postprocessing to eliminate $\nabla p$ (Ku et al., 1987, eq. (15)) together with the neural Clifford layers (Brandstetter et al., 2023), we add a Helmholtz layer $S$ after each application of $\sigma_j \circ (W_j + K_j)$. $S$ performs a discrete Helmholtz decomposition (Girault & Raviart, 2012, Chapter 1 §1 Section 3.1) in the frequency domain to project the latent fields to be solenoidal.

**Difference with Fourier Neural Operator 3d** In view of (2.5), the FFTs in FNO3d transforms are continuous integrals in the spatial dimensions yet a discrete sum in the temporal dimension due to the channel treatment. In the original FNO3d architecture, despite that no explicit restriction imposed on the output time steps, the $d_{\text{out}}$ cannot be trivially changed as the data are prepared by applying a pointwise Gaussian normalizer that depends on $d_{\text{out}}$. Denote the lifting operator (channel expansion) in FNO3d by $P : \{\mathbf{a}_h\} \oplus \{\mathbf{p}_h\} \to \mathbb{R}^{d_v \times d_{\text{out}} \times n \times n}$. Before the application of $P$ in FNO3d, $\mathbf{a}_h$ is artificially repeated $d_{\text{out}}$ times, then $P$ is in a space of linear operators $\simeq \mathbb{R}^{(d_{\text{in}}+3) \times d_v}$. This observation makes it independent of the spatial discretization size $n \times n$ by dependent on $d_{\text{in}}$. In FNO3d, the channel reduction is a pointwise nonlinear universal approximator, yet in ST-FNO3d, this is linear, which eventually facilitate the convexity of the fine-tuning optimization problem.

### D.3 FINE-TUNING

In implementations, we choose $\texttt{Iter}_{\max} = 100$ and the ADAM optimizer with learning rate $10^{-1}$ in Algorithm 1, i.e., an ADAM is simply run for 100 iterations to update parameters that count only a fraction of a spectral layer since the number of parameters accounts for only a single channel.

Table 5: The detailed comparison of FNO3d and its spatiotemporal modification. Layer: # of spectral convolution layers; ST-FNO has an extra layer of spectral convolution in a single channel with skip connection. Modes: $(\tau_{\max}, k_{\max})$. Pre-norm: whether a pointwise Gaussian normalizer is applied for the input data. Eval FLOPs: Giga FLOPs for one evaluation instance. FT: finetuning GFLOPs profiles per ADAM iterations for a $256^2$ grid for a single instance. A `torch.cfloat` type parameter entry counts as two parameters.

| | Architectures | | | | | GFLOPs | | # params |
|---|---|---|---|---|---|---|---|---|
| | layers | channel/width | modes | activation | pre-norm | Eval | FT | |
| FNO3d Example 2 (I) | $4$ | $20$ | $(5, 8)$ | GELU | Y | $13.3$ | N/A | $9.03$m |
| ST-FNO3d Example 2 (I) | $4 + \frac{1}{20}$ | $20$ | $(5, 8)$ | GELU | N | $27.0$ | $3.28$ | $9.02$m |
| FNO3d Example 2 (II) | $4$ | $10$ | $(5, 32)$ | GELU | Y | $28.7$ | N/A | $16.38$m |
| ST-FNO3d Example 2 (II) | $4 + \frac{1}{10}$ | $10$ | $(5, 32)$ | GELU | N | $42.5$ | $3.4$ | $16.42$m |

Table 6: The FLOPs/runtime comparison. FNO3d with a roll-out prediction (10 steps to predict next 10), ST-FNO3d (spatiotemporal prediction), and a traditional numerical solver's time marching (IMEX 4-th order Runge-Kutta). Spatial grid size $256 \times 256$, prediction time interval from $t = 4.5$ to $t = 5.5$ (stable energy cascade regime), numerical solver $\Delta t = 10^{-3}$, roll-out $\delta t = 10\Delta t$, ST-FNO3d $\delta t = 25\Delta t$. Note that ST-FNO3d gets all $40$ steps in a single evaluation, the numerical solver has to march $1000$ steps, and the roll-out prediction inference is supposed to march $10$ times yet becomes de-correlated around $t \approx 4.8$ after the third step. Runtimes are reported for a single instance averaging $20$ runs with a 100w GPU power limiter on a single RTX 4500.

| | Architectures | | | GFLOPs | | | Runtime ($\times 10^{-2}$ s) | | # params |
|---|---|---|---|---|---|---|---|---|---|
| | layers | channel | modes | Eval/Step | FT | All | Single eval | All | |
| FNO3d roll-out Example 2 | $4$ | $10$ | $(5, 32)$ | $28.7$ | N/A | $2.9$M | $4.1 \pm 1.5$ | $45.6 \pm 6.7$ | $16.38$m |
| ST-FNO3d Example 2 | $4 + \frac{1}{10}$ | $10$ | $(5, 32)$ | $1.06$ | $2.51$ | $337.6$ | $22.7 \pm 4.0$ | $162 \pm 4.1$ | $16.42$m |
| IMEX Runge-Kutta | N/A | N/A | N/A | $3.98$ | N/A | $4$M | $0.449 \pm 0.3$ | $450 \pm 8.7$ | $10$ |

**To train or to stop? Investigating the frequency domain signatures**   In Stuart's paper on the convergence of linear operator learning (de Hoop et al., 2023), the evaluation error scales with number of samples, in the few-data regime, the overfitting comes fast. Part of the reason is that the operator learning problems are in the "small" data regime.

Denote $\omega(t, \boldsymbol{x}) := \nabla \times \boldsymbol{u}$, and $\hat{\omega} := \mathcal{F}\omega$ where $\mathcal{F}$ is applied only in space, then we can define enstrophy spectra as follows:

$$\mathcal{E}(t, k) = \sum_{k - \delta k \leq |\boldsymbol{k}| \leq k + \delta k} |\hat{\omega}(t, \boldsymbol{k})|^2, \text{ and } \sum_k \mathcal{E}(t, k) := \int_\Omega |\omega|^2 \mathrm{d}\boldsymbol{x}$$

In Example (II), the enstrophy spectrum decays as $\mathcal{O}(k^{-3})$, which is slightly faster than kinetic energy. This is usually referred to as the direct cascade, as opposed to the inverse cascade $\mathcal{O}(k^{-5/3})$ discovered by Kolmogorov. For ST-FNO, it learns the frequency signature of the data after a single epoch. See Figure 8 and Figure 9.

## D.4   ADDITIONAL EXPERIMENTS

In this subsection, we perform the following additional experiments regarding the possibility of fine-tuning ST-FNO3d for "out-of-distribution" data. All ST-FNO3d models are trained for 15 epochs (5 more than the model in Table 3) with the data generated using the McWilliams's isotropic turbulence data from Example (II) with $\nu = 10^{-3}$. The initial energy distribution satisfies:

$$|\hat{\psi}(k)|^2 \sim k^{-1}(\tau^2 + (k/k_0)^4)^{-1}. \tag{D.1}$$

The parameter $k_0$ in Example (II) roughly means that the initial energy is concentrated at $|\boldsymbol{k}| \approx k_0$. In Table 3 $k_0$ is chosen as $4$ according to McWilliams (1984). For all examples, the same "burn-in"

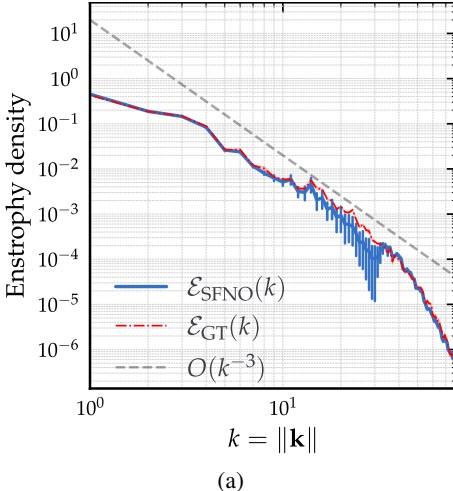 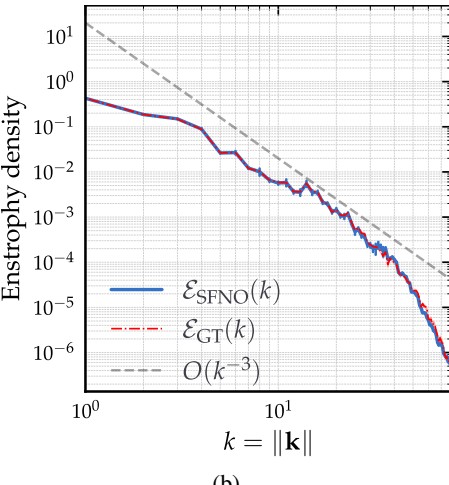

(a)          (b)

Figure 8: Enstrophy spectrum density comparison for Example (II) to illustrate the "convergence" of the ST-FNO training. The ST-FNO evaluation is on a $256 \times 256$ grid for a fixed randomly chosen sample, and the training is on a $64 \times 64$ grid starting from 10 different seeds at different time steps. The error bars are plotted with $+/- 10$ times the standard deviation from the mean to boost the visibility of the convergence. (a): the comparison after 1 epoch, the average relative $L^2$ difference with the ground truth is $2.150 \times 10^{-1}$ ; (b): the comparison after 10 epochs, the average relative $L^2$ difference with the ground truth is $8.051 \times 10^{-2}$.

time is used ($t = 4.5$), the training output time steps are 10 and evaluation output time steps are 40. Please refer to Figure 10 for sample trajectories and Figure 11 for the energy cascade after the "burn-in" time period.

(Ex1) The initial energy is concentrated at $k_0 = 2$. A sample trajectory can be found in Figure 10(a).

(Ex2) Same with above, $k_0 = 8$. A sample trajectory can be found in Figure 10(c).

(Ex3) $k_0 = 4$, $\nu = 1/Re = 1/5000$. In order to resolve the small viscosity, the mesh size becomes $512 \times 512$.

(Ex4) $k_0 = 8$, $\nu = 1/Re = 1/10000$, mesh size is $1024 \times 1024$ for the same reason as above.

The evaluation and fine-tuning result can be found in Table 7. We found that the new paradigm works reasonably well for first three example, yet fails to converge for the last example. For an example plot of the prediction comparison please refer to Figure 12. The correlation graph between the ST-FNO3d's predictions and the ground truth can be found in Figure 13, in which the FNO(2+1)d is adapted from the original FNO2d code. This roll-out-based model predicts the next snapshot based on 10 given snapshots as input, and the predicted snapshot is concatenated to the current input as the latest snapshot to form the input for the next prediction.

Table 7: Evaluation metrics of the McWilliams isotropic turbulence example for "out-of-distribution" data. For details please refer to Section D.4 $\varepsilon := \omega_S - \omega_N$. $r := R(\omega_N)$. $\mathcal{Y} := H^{-1}(\mathbb{T}^2)$. All error norms are evaluated at the final time step. The training is under the $L^2$-norm and the fine-tuning is under the $H^{-1}$-norm. Symbol "–" means the errors are the same as the entry on the row above.

| | In-distribution evaluation | | Out-of-distribution evaluation | | After fine-tuning | |
|---|---|---|---|---|---|---|
| | $\|\varepsilon\|_{L^2}$ | $\|R\|_{-1,n}$ | $\|\varepsilon\|_{L^2}$ | $\|R\|_{-1,n}$ | $\|\varepsilon\|_{L^2}$ | $\|R\|_{-1,n}$ |
| (Ex1) $k_0 = 2$ | $5.39 \times 10^{-2}$ | $1.66 \times 10^{-2}$ | $8.52 \times 10^{-2}$ | $4.25 \times 10^{-2}$ | $2.73 \times 10^{-2}$ | $5.05 \times 10^{-5}$ |
| (Ex2) $k_0 = 8$ | – | – | $1.46 \times 10^{-1}$ | $2.31 \times 10^{-2}$ | $2.06 \times 10^{-2}$ | $5.85 \times 10^{-6}$ |
| (Ex3) $Re = 5 \cdot 10^3$ | – | – | $2.94 \times 10^{-1}$ | $2.73 \times 10^{-2}$ | $2.50 \times 10^{-2}$ | $2.96 \times 10^{-4}$ |
| (Ex4) $Re = 10^4$ | – | – | $4.92 \times 10^{-1}$ | $1.76 \times 10^{-1}$ | $2.355 \times 10^6$ | $2.873 \times 10^4$ |

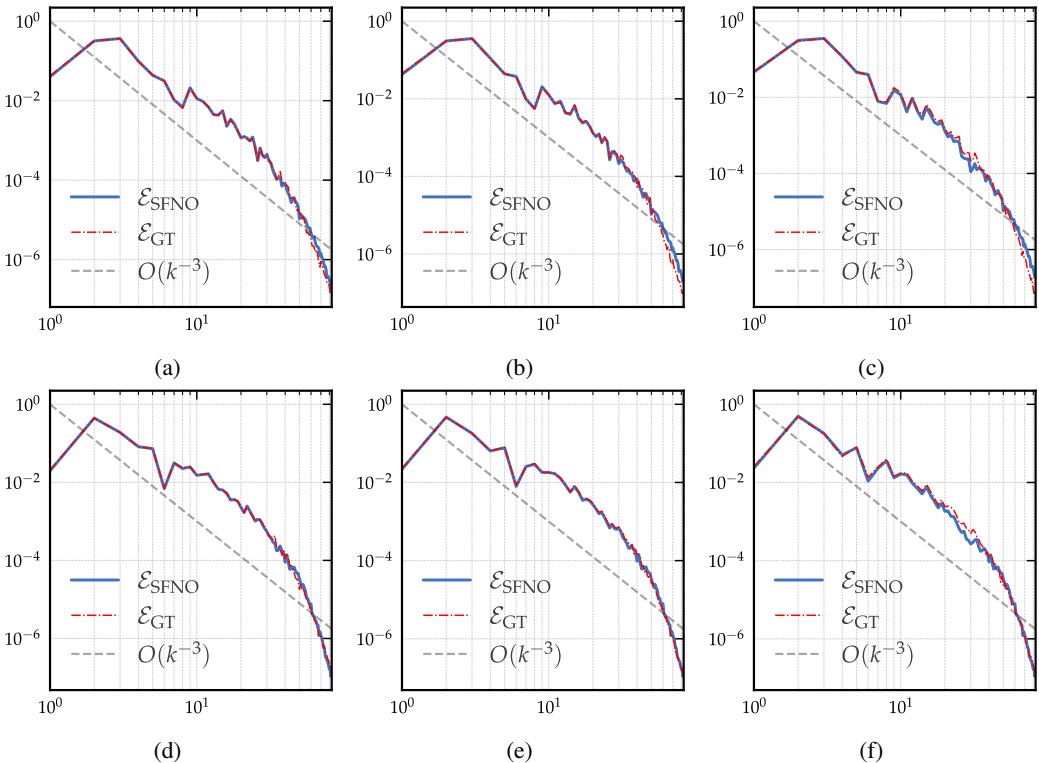

Figure 9: Enstrophy spectrum density comparison for Example (II) for two randomly selected trajectories for a given trained ST-FNO. Training and evaluation setups are identical to those in Figure 8. (a), (b), (c): the comparison of the spectra of the ST-FNO evaluation and the ground truth after 10 epochs of training at $t_{\ell+4}$, $t_{\ell+7}$, and $t_{\ell+10}$ of the first trajectory; (d), (e), (f): the comparison for the second trajectory at the same time steps.

# E ASSUMPTIONS AND PROOFS

**Assumption E.1** (Assumptions for Theorems 3.1, 3.2, E.9). *The following notions and assumptions are adopted throughout the proof of the three theorems involved, all of which are common in the literature for NSE. While some of them can be proved, we opt for list them here to facilitate the presentation.*

($E_1$) *The compact embeddings for Gelfand triple $\mathcal{V} \Subset \mathcal{H} \Subset \mathcal{V}'$ hold, where $\mathcal{V} = \boldsymbol{H}^1(\mathbb{T}^2)$, and $\mathcal{H} = \boldsymbol{L}^2(\mathbb{T}^2)$.*

($E_2$) *The time interval $\mathcal{T}$ in consideration is fixed in that we consider the "refining" problem for a fixed time interval but with variable time steps.*

($E_3$) *The initial condition $\boldsymbol{u}_0 \in \mathcal{V}$, and $\boldsymbol{f} \in L^2(\mathcal{T}; \mathcal{V}')$.*

**Lemma E.2** (Skew-symmetry of the trilinear term). *For $\boldsymbol{z}, \boldsymbol{u}, \boldsymbol{v} \in \boldsymbol{H}^1(\mathbb{T}^2)$, where $\mathbb{T}^2$ denotes $\Omega := (0,1)^2$ equipped with component-wise PBC. Denote*

$$c(\boldsymbol{z}, \boldsymbol{u}, \boldsymbol{v}) := \big((\boldsymbol{z} \cdot \nabla)\boldsymbol{u}, \boldsymbol{v}\big) = \int_\Omega \big((\boldsymbol{z} \cdot \nabla)\boldsymbol{u}\big) \cdot \boldsymbol{v} \, \mathrm{d}\boldsymbol{x}.$$

*If $\nabla \cdot \boldsymbol{z} = 0$, then*

$$c(\boldsymbol{z}, \boldsymbol{u}, \boldsymbol{v}) = -c(\boldsymbol{z}, \boldsymbol{v}, \boldsymbol{u}), \tag{E.1}$$

*and specifically $c(\boldsymbol{z}, \boldsymbol{v}, \boldsymbol{v}) = 0$.*

*Proof.* We note that this result is common in NSE literature, see e.g., Temam (1995, Part I Sec 2.3), for homogeneous boundary or whole space. In this case, it suffices to verify that the boundary term

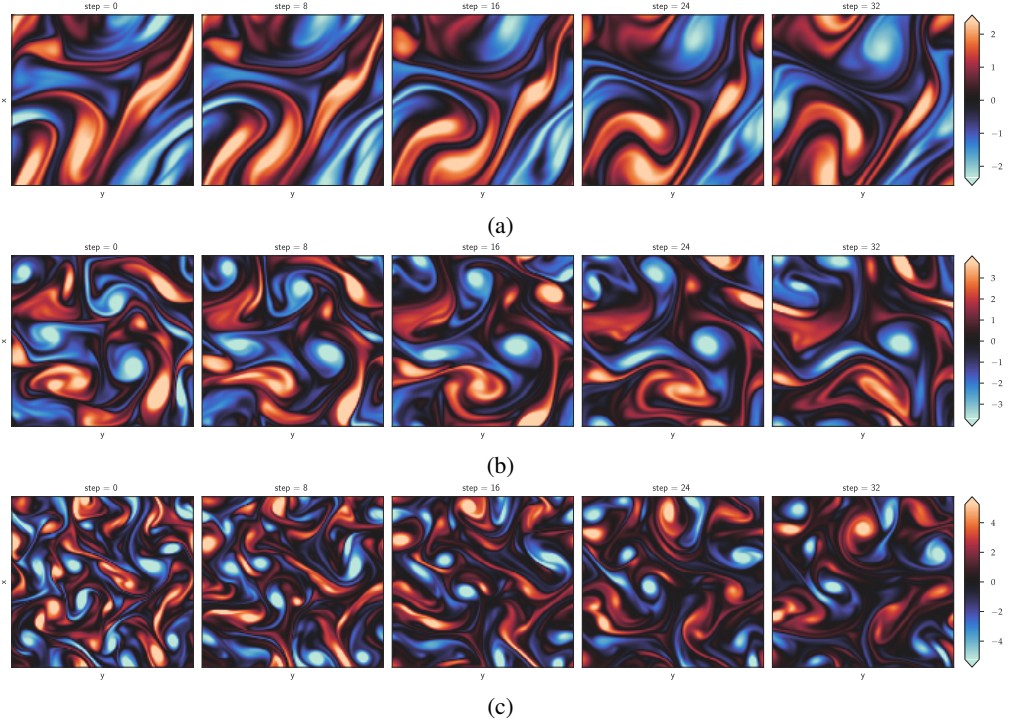

Figure 10: Examples trajectories of the evaluation datasets for additional experiments for Example (II). (a): $k_0 = 2$; (b): $k_0 = 4$. (c): $k_0 = 8$.

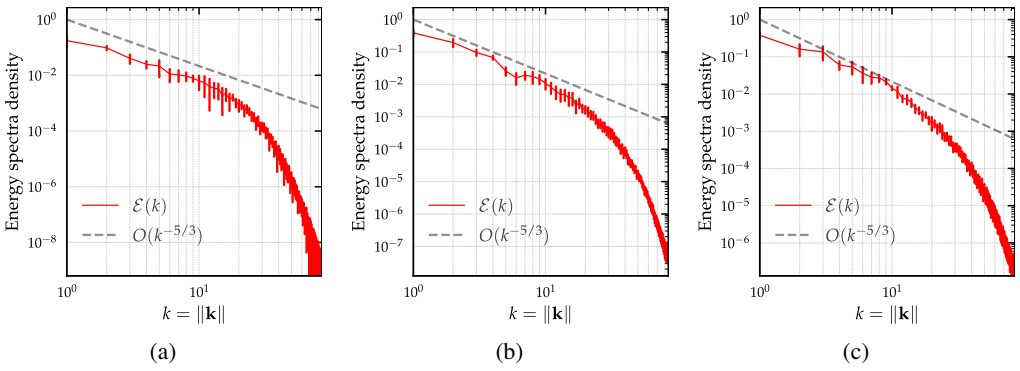

Figure 11: Energy spectrum densities datasets for additional experiments for Example (II). (a): $k_0 = 2$; (b): $k_0 = 4$. (c): $k_0 = 8$.

vanishes. Here we still provide a short argument for the convenience of readers. First by the product rule, and a vector calculus identity of $\nabla \cdot (\boldsymbol{v} \cdot \boldsymbol{u})$, see e.g., Balanis (2012, Appendix II.3.2).

$$\nabla \cdot ((\boldsymbol{v} \cdot \boldsymbol{u})\boldsymbol{z}) = ((\boldsymbol{z} \cdot \nabla)\boldsymbol{u}) \cdot \boldsymbol{v} + ((\boldsymbol{z} \cdot \nabla)\boldsymbol{v}) \cdot \boldsymbol{u} + (\boldsymbol{v} \cdot \boldsymbol{u})\nabla \cdot \boldsymbol{z}. \tag{E.2}$$

By Gauss divergence theorem, we then obtain:

$$\int_\Omega \nabla \cdot ((\boldsymbol{v} \cdot \boldsymbol{u})\boldsymbol{z}) \, \mathrm{d}\boldsymbol{x} = \int_{\partial\Omega} (\boldsymbol{v} \cdot \boldsymbol{u})\boldsymbol{z} \cdot \boldsymbol{n} \, \mathrm{d}s = \sum_{e_i \subset \partial\Omega, 1 \le i \le 4} \int_{\partial\Omega} (\boldsymbol{v} \cdot \boldsymbol{u})\boldsymbol{z} \cdot \boldsymbol{n}_i \, \mathrm{d}s,$$

with $\boldsymbol{n}$ is outer normal with respect to $\partial\Omega$, and $\boldsymbol{n}_i$ denotes that with respect to edge $e_i$. Now on $e_1 := \{x = 0\} \times \{y \in (0, 1)\}$, $\boldsymbol{n}_1 = (-1, 0)^\top$, $\boldsymbol{w}|_{e_1} = \boldsymbol{w}|_{e_2}$ by PBC where $e_2 := \{x = 1\} \times \{y \in (0, 1)\}$ and $\boldsymbol{w} \in \{\boldsymbol{z}, \boldsymbol{u}, \boldsymbol{v}\}$. The integrals on $e_1$ and $e_2$ cancel with one another since $\boldsymbol{n}_2 = -\boldsymbol{n}_1$. Furthermore, if $\nabla \cdot \boldsymbol{z} = 0$, integrating (E.2) on $\Omega$ yields the desired result. $\square$

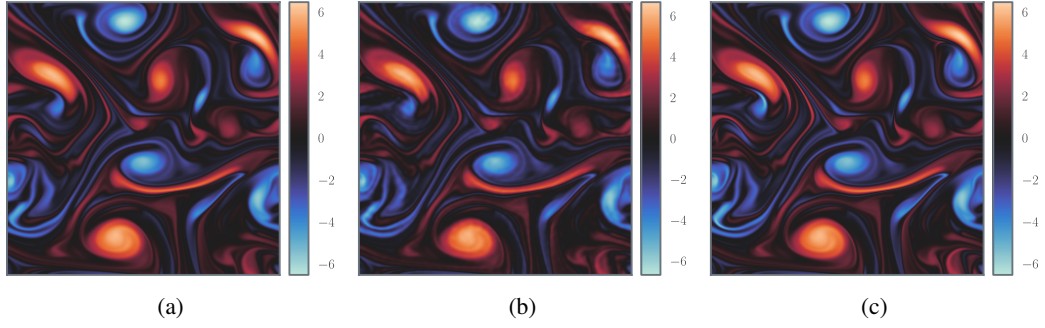

|     |     |     |
| :-: | :-: | :-: |
| (a) | (b) | (c) |

Figure 12: The ground truth versus the ST-FNO3d evaluation/fine-tuning results for the $Re = 5000$ additional experiment for Example (II). (a): the ground truth at $t = 5$ (prediction step 20) generated by IMEX RK4; (b): ST-FNO3d prediction. (c): ST-FNO3d fine-tuned prediction.

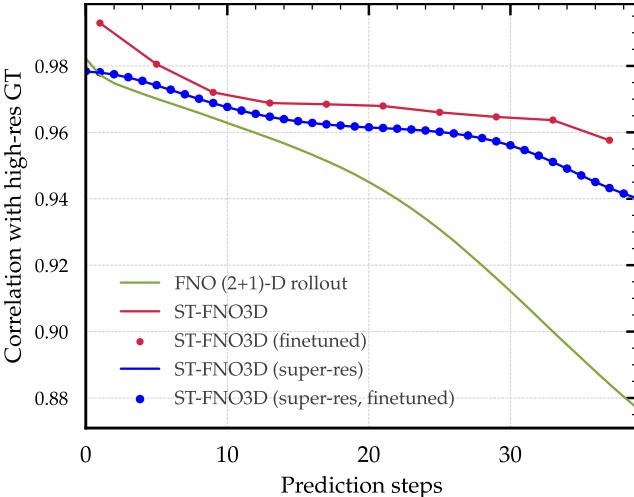

Figure 13: Comparison of the correlation with the ground truth (on $1024 \times 1024$ grid) generated by IMEX RK4 solver.

**Lemma E.3** (Poincaré inequality). *$\mathcal{V}/\mathbb{R} \Subset \mathcal{H}/\mathbb{R}$ is compact and the following Poincaré inequality holds with constant 1: for any $v \in \mathcal{V}/\mathbb{R}$*

$$\|v\| \leq |v|_1 \tag{E.3}$$

*Proof.* A common textbook proof exploits the equivalence of sequential compactness and compactness, once the norm topology is introduced, we prove this in a very intuitive way under our special setting: by definition (3.7), if $v \in \mathcal{V}/\mathbb{R}$ then $\hat{v}(0,0) = 0$

$$\|v\|_0^2 = \sum_{\boldsymbol{k} \in \mathbb{Z}^2 \setminus \{\boldsymbol{0}\}} |\hat{v}(\boldsymbol{k})|^2 \leq \sum_{\boldsymbol{k} \in \mathbb{Z}^2 \setminus \{\boldsymbol{0}\}} |\boldsymbol{k}|^2 |\hat{v}(\boldsymbol{k})|^2 = |v|_1^2.$$

$\square$

### E.1 Proof of Theorem 3.1

To prove Theorem 3.1, we need Lemmas E.4, E.5, and E.6, all of which can be found in classical textbook on analysis and approximations of NSE, such as Temam (Temam, 1995), or Girault & Raviart (Girault & Raviart, 2012) for homogeneous Dirichlet boundary condition for the velocity field or slip boundary condition for the (pseudo-)stress tensor. We shall complement some proofs with simple adaptations to PBC whenever necessary for the convenience of readers.

Before presenting any lemmas, we need the following definitions, a weighted $\boldsymbol{H}^1$-norm on $\mathcal{V} = \boldsymbol{H}^1(\mathbb{T}^2)$ for $\alpha > 0$ on $\Omega$ is defined as

$$\|\boldsymbol{v}\|_{\alpha,1} := \left( \|\boldsymbol{v}\|_{L^2}^2 + \|\alpha\nabla\boldsymbol{v}\|_{L^2}^2 \right)^{1/2}, \text{ and } |\boldsymbol{v}|_{\alpha,1} := \|\alpha\nabla\boldsymbol{v}\|_{L^2}.$$

For $\boldsymbol{f} \in \mathcal{V}'$, define a weighted dual norm as follows:

$$\|\boldsymbol{f}\|_{\alpha^{-1},-1} := \sup_{\boldsymbol{v}\in\mathcal{V}} \frac{\langle\boldsymbol{f},\boldsymbol{v}\rangle}{\|\boldsymbol{v}\|_{\alpha,1}} \tag{E.4}$$

**Lemma E.4** (Contuinity and embedding results for the convection term)**.** *For trilinear convection term*

$$\big((\boldsymbol{u}\cdot\nabla)\boldsymbol{v},\boldsymbol{w}\big) \leq \|\boldsymbol{u}\|_{L^4}\,|\boldsymbol{v}|_1\|\boldsymbol{v}\|_{L^4}. \quad and \quad \big((\boldsymbol{u}\cdot\nabla)\boldsymbol{v},\boldsymbol{w}\big) \leq |\boldsymbol{u}|_1\,|\boldsymbol{v}|_1|\boldsymbol{w}|_1 \tag{E.5}$$

*Proof.* The first result is obtained by the Hölder inequality, and the second holds with suite Sobolev embedding results (Temam, 1995, Part I, §2.3, Lemma 2.1). $\qquad\square$

**Lemma E.5** (Energy stability of NSE in a bounded domain)**.**

$$\|\boldsymbol{u}\|_{L^\infty(\mathcal{T};\mathcal{H})}^2 \leq \|\boldsymbol{u}_0\|_{\mathcal{H}}^2 + \|\boldsymbol{f}\|_{L^2(\mathcal{T};\mathcal{V}')}^2. \tag{E.6}$$

*Proof.* (E.17) is a classical result, see e.g., Temam (1995, Section 3.1). $\qquad\square$

**Lemma E.6** (Fréchet derivative of the convection term)**.** *Given $\boldsymbol{v},\boldsymbol{u} \in \mathcal{V}$, the linearization of the difference of the convection term reads*

$$\boldsymbol{v}\cdot\nabla\boldsymbol{v} = \boldsymbol{u}\cdot\nabla\boldsymbol{u} + D(\boldsymbol{u})(\boldsymbol{v}-\boldsymbol{u}) + \boldsymbol{r} \text{ where } D(\boldsymbol{u})\boldsymbol{\xi} := \big(\boldsymbol{\xi}\cdot\nabla\big)\boldsymbol{u} + (\boldsymbol{u}\cdot\nabla)\boldsymbol{\xi}, \tag{E.7}$$

*is the Fréchet derivative and $\|\boldsymbol{r}\| \lesssim \|\boldsymbol{u}-\boldsymbol{v}\|\,\|\boldsymbol{u}-\boldsymbol{v}\|_1$.*

*Proof.* This result is normally used without proof in linearizing NSE, see e.g., Girault & Raviart (2012, Chapter 4 eq. (6.5.2)). Define $F(\boldsymbol{u},\boldsymbol{G}) := \boldsymbol{u}^\top\boldsymbol{G}$, then expanding $F$ at $\boldsymbol{v}$ and $\boldsymbol{G} = \nabla\boldsymbol{u}$ yields the desired result. $\qquad\square$

**Theorem 2.1 Part I** (Functional-type a posterior error estimate is efficient (rigorous version))**.** *Consider the Gelfand triple $\mathcal{V} \Subset \mathcal{H} \Subset \mathcal{V}'$, and the weak solution $\boldsymbol{u} \in L^2(\mathcal{T};\mathcal{V}) \cap L^\infty(\mathcal{T};\mathcal{H})$ to (2.2) be sufficiently regular, then the dual norm of the residual is efficient to estimate the error:*

$$\|\mathcal{R}(\boldsymbol{u}_\mathcal{N})\|_{L^2(\mathcal{T};\mathcal{V}')}^2 \lesssim \|\partial_t(\boldsymbol{u}-\boldsymbol{u}_\mathcal{N})\|_{L^2(\mathcal{T};\mathcal{V}')}^2 + \|\boldsymbol{u}-\boldsymbol{u}_\mathcal{N}\|_{L^2(\mathcal{T};\mathcal{V})}^2. \tag{E.8}$$

*The constant depend on the regularity of the true solution $\boldsymbol{u}$.*

*Proof.* The proof of the lower bound (efficiency) (E.8) follows the skeleton laid out in Verfürth (2003, Lemma 4.1) for diffusion and temporal derivative terms, while not needing to extend approximation using an affine linear extension operator as Verfürth (2003) does. The treatment of the nonlinear convection term follows Fischer (2015, Lemma 7) for the steady-state NSE (no temporal derivatives), barring the technicality of the divergence-free condition. Consider at $t \in \mathcal{T}$, for any test function $\boldsymbol{v} \in \boldsymbol{H}^1(\mathbb{T}^2)$ and $\nabla\cdot\boldsymbol{v} = 0$, integrating $\boldsymbol{v}$ against $R(\boldsymbol{u}_\mathcal{N})$ in (3.2) yields

$$\begin{aligned}
\langle R(\boldsymbol{u}_\mathcal{N}),\boldsymbol{v}\rangle =& \big(\boldsymbol{f} - \partial_t\boldsymbol{u}_\mathcal{N} - (\boldsymbol{u}_\mathcal{N}\cdot\nabla)\boldsymbol{u}_\mathcal{N} + \nu\Delta\boldsymbol{u}_\mathcal{N},\boldsymbol{v}\big) \\
=& \big(\partial_t(\boldsymbol{u}-\boldsymbol{u}_\mathcal{N}),\boldsymbol{v}\big) - \big(\nu\Delta(\boldsymbol{u}-\boldsymbol{u}_\mathcal{N}),\boldsymbol{v}\big) \\
&+ \big((\boldsymbol{u}\cdot\nabla)\boldsymbol{u} - (\boldsymbol{u}_\mathcal{N}\cdot\nabla)\boldsymbol{u}_\mathcal{N},\boldsymbol{v}\big).
\end{aligned}$$

Integrating by parts for the diffusion term, using the same argument for PBC as in Lemma E.2, and inserting $(\boldsymbol{u}\cdot\nabla)\boldsymbol{u}_\mathcal{N}$ yield

$$\langle R(\boldsymbol{u}_\mathcal{N}),\boldsymbol{v}\rangle = \underbrace{\big(\partial_t(\boldsymbol{u}-\boldsymbol{u}_\mathcal{N}),\boldsymbol{v}\big)}_{(\mathrm{I})} + \underbrace{\big(\nu\nabla(\boldsymbol{u}-\boldsymbol{u}_\mathcal{N}),\nabla\boldsymbol{v}\big)}_{(\mathrm{II})} \\ + \underbrace{\big((\boldsymbol{u}\cdot\nabla)(\boldsymbol{u}-\boldsymbol{u}_\mathcal{N}),\boldsymbol{v}\big)}_{(\mathrm{III})} + \underbrace{\big(((\boldsymbol{u}-\boldsymbol{u}_\mathcal{N})\cdot\nabla)\boldsymbol{u}_\mathcal{N},\boldsymbol{v}\big)}_{(\mathrm{IV})}. \tag{E.9}$$

Applying the definition of a weighted dual norm (E.4) on (I) we have

$$(\mathrm{I}) \le \|\partial_t(\boldsymbol{u} - \boldsymbol{u}_\mathcal{N})(t,\cdot)\|_{\nu^{-1/2},-1} \|\boldsymbol{v}\|_{\nu^{1/2},1}.$$

For (II) we simply have

$$(\mathrm{II}) \le \|\nu^{1/2}\nabla(\boldsymbol{u} - \boldsymbol{u}_\mathcal{N})\|_{L^2} \|\nu^{1/2}\nabla\boldsymbol{v}\|_{L^2}.$$

For (III), applying Lemma E.4 and the stability estimate in Lemma E.5 for $\boldsymbol{u}$:

$$(\mathrm{III}) \le C\|\boldsymbol{u}\|_{\nu^{-1},1}\|\boldsymbol{u} - \boldsymbol{u}_\mathcal{N}\|_{\nu^{1/2},1}\|\boldsymbol{v}\|_{\nu^{1/2},1} \le \nu^{-1}C_1(t,\boldsymbol{u}_0)\|\boldsymbol{u} - \boldsymbol{u}_\mathcal{N}\|_{\nu^{1/2},1}\|\boldsymbol{v}\|_{\nu^{1/2},1}.$$

For (IV), applying Lemma E.4 and the stability estimate in Lemma E.5 for $\boldsymbol{u}_\mathcal{N}$:

$$(\mathrm{IV}) \le C\|\boldsymbol{u} - \boldsymbol{u}_\mathcal{N}\|_{\nu^{1/2},1}\|\boldsymbol{u}_\mathcal{N}\|_{\nu^{-1},1}\|\boldsymbol{v}\|_{\nu^{1/2},1} \le \nu^{-1}C_2(t,\boldsymbol{u}_0)\|\boldsymbol{u} - \boldsymbol{u}_\mathcal{N}\|_{\nu^{1/2},1}\|\boldsymbol{v}\|_{\nu^{1/2},1}.$$

Applying the Cauchy-Schwarz inequality, one has

$$|\langle R(\boldsymbol{u}_\mathcal{N}), \boldsymbol{v}\rangle|^2 \le C\left(\|\partial_t(\boldsymbol{u} - \boldsymbol{u}_\mathcal{N})(t,\cdot)\|_{\nu^{-1/2},-1}^2 + \|\boldsymbol{u} - \boldsymbol{u}_\mathcal{N}\|_{\nu^{1/2},1}^2\right)\|\boldsymbol{v}\|_{\nu^{1/2},1}^2.$$

Finally, equipping $\mathcal{V}'$ with the weighted norm (E.4), and by the definition of $L^2(\mathcal{T};\mathcal{V}')$, we have

$$\begin{aligned}
\|R(\boldsymbol{u}_\mathcal{N})\|_{L^2(\mathcal{T};\mathcal{V})}^2 &= \int_\mathcal{T} \sup_{\boldsymbol{v}\in\mathcal{V}} \frac{|\langle R(\boldsymbol{u}_\mathcal{N}),\boldsymbol{v}\rangle|^2}{\|\boldsymbol{v}\|_{\nu^{1/2},1}^2}\,\mathrm{d}t \\
&\le C\int_\mathcal{T}\left(\|\partial_t(\boldsymbol{u} - \boldsymbol{u}_\mathcal{N})(t,\cdot)\|_{\nu^{-1/2},-1}^2 + \|\boldsymbol{u} - \boldsymbol{u}_\mathcal{N}\|_{\nu^{1/2},1}^2\right)\mathrm{d}t,
\end{aligned}$$

where $C = C(\boldsymbol{u}_0, \nu, \mathcal{T})$. $\qquad\square$

**Theorem 2.1 Part II** (Functional-type a posterior error estimate is reliable (rigorous version)).
*Consider the Gelfand triple $\mathcal{V} \Subset \mathcal{H} \Subset \mathcal{V}'$, and the weak solution is $\boldsymbol{u} \in L^2(\mathcal{T};\mathcal{V}) \cap L^\infty(\mathcal{T};\mathcal{H})$ to (2.2). We assume that*

(2.1.1) $\boldsymbol{u}_\mathcal{N}$ *and $\boldsymbol{u}$ are sufficiently close in the sense that: there exists $\gamma \in [0, C)$ for a fixed $C \in \mathbb{R}^+$, for $\boldsymbol{J}$ defined in Lemma E.6*

$$\|\boldsymbol{J}(\boldsymbol{u}, \boldsymbol{u}_\mathcal{N})\|_{\mathcal{V}'} \le \gamma\|\nabla(\boldsymbol{u} - \boldsymbol{u}_\mathcal{N})\|. \tag{E.10}$$

(2.1.2) $\boldsymbol{u}$ *satisfies the Gårding inequality (a weaker coercivity): for any $\boldsymbol{v}(t,\cdot) \in \mathcal{V}$, define*

$$B(\boldsymbol{v}, \boldsymbol{w}; \boldsymbol{u}) := (\partial_t\boldsymbol{v}, \boldsymbol{w}) + \nu(\nabla\boldsymbol{v}, \nabla\boldsymbol{w}) + ((\boldsymbol{u}\cdot\nabla)\boldsymbol{v}, \boldsymbol{w}),$$

*and for $\boldsymbol{v}$ in a neighborhood such that Assumption (2.1.1) holds, there exists $\alpha, \beta > 0$ such that $\alpha - \beta \ge \nu$,*

$$B(\boldsymbol{v}, \boldsymbol{v}; \boldsymbol{u}) + \beta\|\boldsymbol{v}\|^2 \ge \frac{\mathrm{d}}{\mathrm{d}t}\|\boldsymbol{v}\|^2 + \alpha\|\nabla\boldsymbol{v}\|^2. \tag{E.11}$$

*The dual norm of the residual is then reliable to serve as an error measure in the following sense: denote $\mathcal{T}_m := (t_m, t_{m+1}]$*

$$\|\boldsymbol{u} - \boldsymbol{u}_\mathcal{N}\|_{L^\infty(\mathcal{T}_m;\mathcal{H})}^2 + \|\boldsymbol{u} - \boldsymbol{u}_\mathcal{N}\|_{L^2(\mathcal{T}_m;\mathcal{V})}^2 \le \left\|(\boldsymbol{u} - \boldsymbol{u}_\mathcal{N})(t_m,\cdot)\right\|_\mathcal{V}^2 + C\int_{\mathcal{T}_m}\|R(\boldsymbol{u}_\mathcal{N})(t,\cdot)\|_{\mathcal{V}'}^2\,\mathrm{d}t. \tag{E.12}$$

*The constant $C = C(\nu)$.*

*Proof.* The proof of part II of Theorem 3.1 combines the meta-framework of Verfürth (2003) for time-dependent problems and Verfürth (1994, Section 8) for the stationary NSE. Note that thanks for the construction of divergence-free $\boldsymbol{u}_\mathcal{N}$, the technicality of applying the Ladyzhenskaya-Babušška-Brezzi inf-sup condition is avoided. In the meantime, no interpolations are needed to convert the functional norm for the negative Sobolev spaces to an $L^2$ representation.

To prove the upper bound (reliability) (3.5), we simply choose a time step $t = t_m$, and let $\boldsymbol{v} = (\boldsymbol{u} - \boldsymbol{u}_\mathcal{N})(t,\cdot)$ on $(t_m, t_{m+1})$. Note that the discretized approximation's evaluation at $t$, which may

not be the temporal grids $t_m$ or $t_{m+1}$ are naturally defined using the basis in (3.1). Using the error representation in (E.9), we have the (I) and (II) terms in $\langle R(\boldsymbol{u}_\mathcal{N}), \boldsymbol{u} - \boldsymbol{u}_\mathcal{N}\rangle$ are

$$
\begin{aligned}
\text{(I)} + \text{(II)} &= \left(\partial_t(\boldsymbol{u} - \boldsymbol{u}_\mathcal{N}), \boldsymbol{u} - \boldsymbol{u}_\mathcal{N}\right) + \left(\nu\nabla(\boldsymbol{u} - \boldsymbol{u}_\mathcal{N}), \nabla(\boldsymbol{u} - \boldsymbol{u}_\mathcal{N})\right) \\
&= \frac{1}{2}\frac{\mathrm{d}}{\mathrm{d}t}\|\boldsymbol{u} - \boldsymbol{u}_\mathcal{N}\|_{L^2}^2 + \nu\|\nabla(\boldsymbol{u} - \boldsymbol{u}_\mathcal{N})\|_{L^2}^2
\end{aligned}
\tag{E.13}
$$

For the (III) and (IV) terms we have

$$
\text{(III)} + \text{(IV)} = \overbrace{\left((\boldsymbol{u}\cdot\nabla)(\boldsymbol{u} - \boldsymbol{u}_\mathcal{N}), \boldsymbol{u} - \boldsymbol{u}_\mathcal{N}\right)}^{=0 \text{ by Lemma E.2}} + \overbrace{\left(((\boldsymbol{u} - \boldsymbol{u}_\mathcal{N})\cdot\nabla)\boldsymbol{u}_\mathcal{N}, \boldsymbol{u} - \boldsymbol{u}_\mathcal{N}\right)}^{=:(*)}.
\tag{E.14}
$$

Note by Lemma E.2, $(*)$ is also:

$$
(*) = \left(((\boldsymbol{u} - \boldsymbol{u}_\mathcal{N})\cdot\nabla)\boldsymbol{u}, \boldsymbol{u} - \boldsymbol{u}_\mathcal{N}\right),
\tag{E.15}
$$

since $c(\boldsymbol{z}, \boldsymbol{z}, \boldsymbol{z}) = 0$ for $\boldsymbol{z} = \boldsymbol{u} - \boldsymbol{u}_\mathcal{N}$. Consequently, by a vector calculus identity, we have

$$
\int_\Omega (\boldsymbol{z}\cdot\nabla)\boldsymbol{u}\cdot\boldsymbol{z}\,\mathrm{d}\boldsymbol{x} = \int_\Omega \nabla\boldsymbol{u} : (\boldsymbol{z}\otimes\boldsymbol{z})\,\mathrm{d}\boldsymbol{x}.
$$

Thus, we reach the following error equation:

$$
\langle R(\boldsymbol{u}_\mathcal{N}), \boldsymbol{u} - \boldsymbol{u}_\mathcal{N}\rangle = \frac{1}{2}\frac{\mathrm{d}}{\mathrm{d}t}\|\boldsymbol{u} - \boldsymbol{u}_\mathcal{N}\|_{L^2}^2 + \nu\|\nabla(\boldsymbol{u} - \boldsymbol{u}_\mathcal{N})\|_{L^2}^2 + \int_\Omega \nabla\boldsymbol{u} : ((\boldsymbol{u} - \boldsymbol{u}_\mathcal{N})\otimes(\boldsymbol{u} - \boldsymbol{u}_\mathcal{N}))\,\mathrm{d}\boldsymbol{x}.
\tag{E.16}
$$

Now, by Assumption (2.1.2), Poincaré inequality from Lemma E.3,

$$
\begin{aligned}
\frac{1}{2}\frac{\mathrm{d}}{\mathrm{d}t}\|\boldsymbol{u} - \boldsymbol{u}_\mathcal{N}\|^2 + \nu\|\nabla(\boldsymbol{u} - \boldsymbol{u}_\mathcal{N})\|^2 &\leq \frac{\mathrm{d}}{\mathrm{d}t}\|\boldsymbol{u} - \boldsymbol{u}_\mathcal{N}\|^2 + (\alpha - \beta)\|\nabla(\boldsymbol{u} - \boldsymbol{u}_\mathcal{N})\|^2 \\
&\leq \frac{\mathrm{d}}{\mathrm{d}t}\|\boldsymbol{u} - \boldsymbol{u}_\mathcal{N}\|^2 + \alpha\|\nabla(\boldsymbol{u} - \boldsymbol{u}_\mathcal{N})\|^2 - \beta\|\boldsymbol{u} - \boldsymbol{u}_\mathcal{N}\| \\
&\leq B(\boldsymbol{u} - \boldsymbol{u}_\mathcal{N}, \boldsymbol{u} - \boldsymbol{u}_\mathcal{N}; \boldsymbol{u}).
\end{aligned}
$$

By (E.13), (E.14), and (E.15),

$$
\begin{aligned}
B(\boldsymbol{u} - \boldsymbol{u}_\mathcal{N}, \boldsymbol{u} - \boldsymbol{u}_\mathcal{N}; \boldsymbol{u}) &= \text{(I)} + \text{(II)} + \text{(III)} + \text{(IV)} = \langle R(\boldsymbol{u}_\mathcal{N}), \boldsymbol{u} - \boldsymbol{u}_\mathcal{N}\rangle \\
&\leq \|R(\boldsymbol{u}_\mathcal{N})\|_{\nu^{-1/2},-1}|\boldsymbol{u} - \boldsymbol{u}_\mathcal{N}|_{\nu^{1/2},1} \\
&\leq \frac{1}{2}\|R(\boldsymbol{u}_\mathcal{N})\|_{\nu^{-1/2},-1}^2 + \frac{1}{2}|\boldsymbol{u} - \boldsymbol{u}_\mathcal{N}|_{\nu^{1/2},1}^2.
\end{aligned}
$$

As a result, we have

$$
\frac{\mathrm{d}}{\mathrm{d}t}\|\boldsymbol{u} - \boldsymbol{u}_\mathcal{N}\|^2 + \nu\|\nabla(\boldsymbol{u} - \boldsymbol{u}_\mathcal{N})\|^2 \leq \|R(\boldsymbol{u}_\mathcal{N})\|_{\nu^{-1/2},-1}^2.
\tag{E.17}
$$

Integrating from $t_m$ to any $t \in (t_m, t_{m+1}]$ yields

$$
\|(\boldsymbol{u} - \boldsymbol{u}_\mathcal{N})(t, \cdot)\|^2 - \|(\boldsymbol{u} - \boldsymbol{u}_\mathcal{N})(t_m, \cdot)\|^2 + \nu\|\nabla(\boldsymbol{u} - \boldsymbol{u}_\mathcal{N})\|_{L^2(t_m, t; \mathcal{H})}^2 \leq \int_{t_m}^t \|R(\boldsymbol{u}_\mathcal{N})(t, \cdot)\|_{\nu^{-1/2},-1}^2\,\mathrm{d}t.
$$

Since $t \in (t_m, t_{m+1}]$ is arbitrary, we have proved the reliability (E.12). $\qquad\square$

**Remark E.7** (Necessity of the "sufficient close" and the regularity conditions). *First, the main hurdle to prove the reliability is that the convection term $(*)$ is not positive definite, one would encounter some difficult in deriving the upper bound as yet moving $(*)$ to the right-hand side in (E.16) does not yield a meaningful estimate,*

$$
\text{norm of the error} \simeq \text{(I)} + \text{(II)} = \langle R(\boldsymbol{u}_\mathcal{N}), \boldsymbol{u} - \boldsymbol{u}_\mathcal{N}\rangle - (*) \nleq \langle R(\boldsymbol{u}_\mathcal{N}), \boldsymbol{u} - \boldsymbol{u}_\mathcal{N}\rangle
$$

*We note that Assumption (2.1.2) is equivalent to putting a threshold on $\|\boldsymbol{u}(t, \cdot)\|_{L^\infty}$, which is commonly assumed in the analysis of numerical methods for convection-dominated problems. A stronger alternative assumption than Assumption (2.1.2) would be imposing a stronger constraint*

*on $\gamma$ in Assumption (2.1.1) Using the "sufficient close" assumption (E.10) on the nonlinear term, we have*

$$|(*)| \leq \| \left( (\boldsymbol{u} - \boldsymbol{u}_\mathcal{N}) \cdot \nabla \right) \boldsymbol{u}_\mathcal{N} \|_{\mathcal{V}'} \| \boldsymbol{u} - \boldsymbol{u}_\mathcal{N} \|_{\mathcal{V}} \leq \gamma \| \nabla (\boldsymbol{u} - \boldsymbol{u}_\mathcal{N}) \|^2.$$

*Now if $\bar{\nu} := \nu - \gamma \geq 0$, one would reach a similar estimate as (E.17) by replacing $\nu$ with $\bar{\nu}$. If no extra assumption on $\gamma$ is imposed.*

*Another way is to consider a Sobolev embedding directly after applying the Hölder inequality, for (E.15) we have*

$$|(*)| \leq \int_\Omega |\nabla \boldsymbol{u}| |\boldsymbol{u} - \boldsymbol{u}_\mathcal{N}|^2 \leq \|\nabla \boldsymbol{u}\| \, \|\boldsymbol{u} - \boldsymbol{u}_\mathcal{N}\|_{L^4}^2 \leq \rho \|\nabla \boldsymbol{u}\| \, \|\nabla (\boldsymbol{u} - \boldsymbol{u}_\mathcal{N})\|^2,$$

*where $\rho$ is the constant in the following Sobolev embedding*

$$\|\boldsymbol{u} - \boldsymbol{u}_\mathcal{N}\|_{L^4} \leq \rho \|\nabla (\boldsymbol{u} - \boldsymbol{u}_\mathcal{N})\|.$$

*Now if $\tilde{\nu} := \nu - \rho \|\nabla \boldsymbol{u}\| \geq 0$, one would reach a similar estimate as (E.17) by replacing $\nu$ with $\tilde{\nu}$.*

### E.2 PROOF OF THEOREM 3.2

**Theorem 2.2** (Equivalence of functional norm and negative Sobolev norm). *Consider the Gelfand triple $\mathcal{V} \Subset \mathcal{H} \Subset \mathcal{V}'$, then*

$$\|f\|_{\mathcal{V}'} = |f|_{-1} \text{ for } f \in \mathcal{H}/\mathbb{R}. \tag{E.18}$$

*Proof.* This proof leverages the spectral basis for $\mathcal{H}$ without the extra technicality of Schwartz space involved if one ought to assume that $f \in \mathcal{V}'$. Recall (3.1), and define an infinite version

$$\mathcal{S}_\infty := \text{span} \left\{ e^{i\boldsymbol{k}\cdot\boldsymbol{x}} : \boldsymbol{k} \in 2\pi\mathbb{Z}^2 \right\}. \tag{E.19}$$

In what follows, we shall omit the extra technicality that one has to work on partial sums first, and then considers the convergence in the corresponding norms. We simply take for granted that the differentiation and integration/sum can be interchanged, and directly identify $f \in \mathcal{H}/\mathbb{R}$ with its Fourier series in $\mathcal{S}_\infty$,

$$f(\boldsymbol{x}) = \sum_{\boldsymbol{k} \in 2\pi\mathbb{Z}^2} \hat{f}(\boldsymbol{k}) e^{i\boldsymbol{k}\cdot\boldsymbol{x}}.$$

Note since $f$ lies in the quotient space, $\hat{f}(0,0) = 0$. Then, the duality pairing can be identified as an $L^2$-inner product, as well as the test function $v \in \mathcal{H}$,

$$\begin{aligned} \langle f, v \rangle_{\mathcal{V}',\mathcal{V}} = (f, v) &= \left( \sum_{\boldsymbol{k} \in 2\pi\mathbb{Z}^2} \hat{f}(\boldsymbol{k}) e^{i\boldsymbol{k}\cdot\boldsymbol{x}}, \sum_{\boldsymbol{m} \in 2\pi\mathbb{Z}^2} \overline{\hat{v}(\boldsymbol{m}) e^{i\boldsymbol{m}\cdot\boldsymbol{x}}} \right) \\ &= \sum_{\boldsymbol{k} \in 2\pi\mathbb{Z}_n^2 \setminus \{0\}} \hat{f}(\boldsymbol{k}) \overline{\hat{v}(\boldsymbol{k})} \\ &\leq \left( \sum_{\boldsymbol{k} \in 2\pi\mathbb{Z}_n^2 \setminus \{0\}} |\boldsymbol{k}|^{-2} |\hat{f}(\boldsymbol{k})|^2 \right)^{1/2} \left( \sum_{\boldsymbol{k} \in 2\pi\mathbb{Z}_n^2 \setminus \{0\}} |\boldsymbol{k}|^2 |\hat{v}(\boldsymbol{k})|^2 \right)^{1/2} \\ &\leq |f|_{-1} |v|_1. \end{aligned}$$

As a result, by the definition in (3.7) and the estimate above

$$\|f\|_{\mathcal{V}'} = \sup_{v \in \mathcal{V}/\mathbb{R}, |v|_{\mathcal{V}}=1} |\langle f, v \rangle| = \sup_{v \in \mathcal{V}/\mathbb{R}} \frac{(f, v)}{|v|_1} \leq |f|_{-1}.$$

To show the other direction, let $v = (-\Delta)^{-1} f \in H^1(\mathbb{T}^2)$ by the well-posedness of $-\Delta v = f$ for $f \in L^2(\mathbb{T}^2)$. We have

$$
\begin{aligned}
\langle f, (-\Delta)^{-1} f \rangle = (-\Delta v, v) &= \left( f, (-\Delta)^{-1} f \right) \\
&= \left( \sum_{\boldsymbol{k} \in 2\pi\mathbb{Z}^2} \hat{f}(\boldsymbol{k}) e^{i\boldsymbol{k}\cdot\boldsymbol{x}}, \sum_{\boldsymbol{m} \in 2\pi\mathbb{Z}^2} (-\Delta_{\boldsymbol{x}})^{-1} \overline{\hat{f}(\boldsymbol{m}) e^{i\boldsymbol{m}\cdot\boldsymbol{x}}} \right) \\
&= \left( \sum_{\boldsymbol{k} \in 2\pi\mathbb{Z}^2} \hat{f}(\boldsymbol{k}) e^{i\boldsymbol{k}\cdot\boldsymbol{x}}, \sum_{\boldsymbol{m} \in 2\pi\mathbb{Z}^2} \frac{1}{|\boldsymbol{m}|^2} \overline{\hat{f}(\boldsymbol{m}) e^{i\boldsymbol{m}\cdot\boldsymbol{x}}} \right) \\
&= \sum_{\boldsymbol{k} \in 2\pi\mathbb{Z}_n^2 \setminus \{0\}} |\boldsymbol{k}|^{-2} |\hat{f}(\boldsymbol{k})|^2 = |f|_{-1}^2.
\end{aligned}
$$

Note that by the construction of $v$ above, $|v|_1 = |f|_{-1}$, thus,

$$
\sup_{v \in \mathcal{V}} \frac{(f, v)}{|v|_1} \geq \frac{\left( f, (-\Delta)^{-1} f \right)}{|v|_1} = |f|_{-1},
$$

which proves the theorem.

$\square$

### E.3 CONVERGENCE RESULT OF FINE-TUNING

In the following theorem, Theorem E.9, we show that a sufficient condition for the optimizer to converge is to "reach" a neighborhood of the true solution, thus corroborating the necessity of training. Note in both Theorems 3.1 and E.9, the error term of $\boldsymbol{u}_{\mathcal{N}}^{(m)} - \boldsymbol{u}(t_m, \cdot)$ is present. We have to acknowledge that the "initial value" for the predicted trajectory, which is the last snapshot in the input trajectory, may have errors. This is the major motivation that we opt to use $\boldsymbol{u}_{\mathcal{N}}^{(\ell)}$, which is the input trajectory's last snapshot in the skip-connection in $\widetilde{Q}$

$$
\boldsymbol{u}_{\mathcal{N}}^{(m+1)} = \boldsymbol{u}_{\mathcal{N}}^{(\ell)} + \widetilde{Q}_\theta(\boldsymbol{v}_{\mathcal{N}}^{(m+1)}), \quad \text{for } m = \ell, \dots, \ell + n_t - 1. \tag{E.20}
$$

In some sense, ST-FNO learns the *derivative* $\partial_t \boldsymbol{u}$'s arbitrary-length integral. If one wants to modify Algorithm 1 in view of Theorem E.9 such that the error control is guaranteed, the following algorithm can be used but loses the "parallel-in-time" nature. We also note that , thanks to the spectral convolution in $\widetilde{Q}$ being affine *linear*, showing Theorem E.9 is quite straightforward, as one has to establish the connection between fine-tuning and seeking a nonlinear Galerkin projection in Fourier space (3.1) under the functional norm. Let $\theta^* = \arg\min_\theta \|\mathcal{R}(\boldsymbol{u}_{\mathcal{N}}(\theta))(t_{m+1}, \cdot)\|_{-1}$, then $\widetilde{Q}_{\theta^*}(\boldsymbol{u}_{\mathcal{N}}) = \arg\min_{\boldsymbol{v} \in \mathcal{S}} \|\boldsymbol{u}(t_{m+1}, \cdot) - (\boldsymbol{u}(t_m, \cdot) + \boldsymbol{v})\|_*$.

**Lemma E.8** (Local strict convexity for the fine-tuning loss). *Define $\| \cdot \|_{*, \delta}$ to be a (dual) graph norm on $L^\infty(\mathcal{T}_\delta; \boldsymbol{L}^2(\mathbb{T}^2)) \cap L^2(\mathcal{T}_\delta; \boldsymbol{H}^1(\mathbb{T}^2))$, where $\mathcal{T}_\delta := [t - \delta, t + \delta]$*

$$
\|\boldsymbol{v}\|_{*, \delta} = \left\{ \fint_{\mathcal{T}_\delta} \|\boldsymbol{v}(t, \cdot)\|^2 \mathrm{d}t + \fint_{\mathcal{T}_\delta} \|(\partial_t \boldsymbol{v} + \boldsymbol{v} \cdot \nabla \boldsymbol{v})(t, \cdot)\|_{\mathcal{V}'}^2 \mathrm{d}t \right\}^{1/2}
$$

*For $\boldsymbol{u} \in L^\infty(\mathcal{T}_\delta; \boldsymbol{L}^2(\mathbb{T}^2)) \cap L^2(\mathcal{T}_\delta; \boldsymbol{H}^1(\mathbb{T}^2))$ the weak solution to (2.2) on $\mathcal{T}_\delta$ that is sufficiently smooth, there exists $\delta, \epsilon \in \mathbb{R}^+$, such that on*

$$
\mathcal{B}(\boldsymbol{u}; \epsilon) := \left\{ \boldsymbol{v} \in L^\infty(\mathcal{T}_\delta; \boldsymbol{L}^2(\mathbb{T}^2)) \cap L^2(\mathcal{T}_\delta; \boldsymbol{H}^1(\mathbb{T}^2)) : \|\boldsymbol{u} - \boldsymbol{v}\|_{*, \delta} \leq \epsilon \right\},
$$

*the functional*

$$
J(\boldsymbol{v}(t, \cdot)) := \frac{1}{2} \|R(\boldsymbol{v})\|_{\mathcal{V}'}^2, \quad \text{where} \ R(\boldsymbol{v}) := \boldsymbol{f} - \partial_t \boldsymbol{v} - (\boldsymbol{v} \cdot \nabla)\boldsymbol{v} + \nu \Delta \boldsymbol{v}
$$

*is strictly convex.*

*Proof.* First, by Theorem 3.2,

$$
J(\boldsymbol{v}) = \frac{1}{2} \left\langle R(\boldsymbol{v}), (-\Delta)^{-1} R(\boldsymbol{v}) \right\rangle_{\mathcal{V}', \mathcal{V}}
$$

---

**Algorithm 2** An error-guarantee fine-tuning strategy.

---

**Input:** ST-FNO $\mathcal{G}_{\theta,\Theta} := \widetilde{Q}_\theta \circ \mathcal{G}_\Theta$; time stepping scheme $B_\gamma(\cdot)$; optimizer $\mathcal{D}(\theta, \nabla_\theta(\cdot))$; training
     dataset: solution trajectories at $[t_1, \ldots, t_{\ell'}]$ as input and at $[t_{\ell'+1}, \ldots, t_{\ell'+n_t'}]$ as output.
  1: Train the ST-FNO model until the energy signature matches the inverse cascade.
  2: Freeze $\Theta$ in $\mathcal{G}_\Theta$ of ST-FNO $\mathcal{G}_{\theta,\Theta}$, keep $\widetilde{Q}_\theta$ trainable.
  3: Cast all nn.Module involved and tensors to torch.float64 and torch.complex128 hereafter.
**Input:** Evaluation dataset: solution trajectories at $[t_1, \ldots, t_\ell]$ as input, output time step $n_t$.
  4: **for** $m = \ell, \cdots, \ell + n_t - 1$ **do**
  5:      Extract the latent fields $\boldsymbol{v}_{\mathcal{N}}^{(m+1)}$ output of $\mathcal{G}_\Theta$ at $t_{m+1}$ and hold them fixed.
  6:      By construction of ST-FNO: such that $\boldsymbol{u}_{\mathcal{N}}^{(m+1)}(\theta) := \boldsymbol{u}_{\mathcal{N}}^{(m)} + \widetilde{Q}_\theta(\boldsymbol{v}_{\mathcal{N}}^{(m+1)})$.
  7:      March one step with $(\Delta t)^\gamma$ using $B_\gamma$: $D_t \boldsymbol{u}_{\mathcal{N}}(\theta) := (\Delta t)^{-\gamma}(B_\gamma(\boldsymbol{u}_{\mathcal{N}}(\theta)) - \boldsymbol{u}_{\mathcal{N}}(\theta))$.
  8:      $j \leftarrow 0$
  9:      **while** $\eta_m(\boldsymbol{u}_{\mathcal{N}}(\theta), D_t \boldsymbol{u}_{\mathcal{N}}(\theta)) > \text{Tol}$ **do**
 10:         Apply the optimizer to update parameters in $\widetilde{Q}$: $\theta \leftarrow \mathcal{D}(\theta, \nabla_\theta(\eta_m^2))$, $j \leftarrow j + 1$.
 11:         Forward pass only through $\widetilde{Q}$ to update $\boldsymbol{u}_{\mathcal{N}} \leftarrow \boldsymbol{u}_{\mathcal{N}}^{(m)} + \widetilde{Q}_\theta(\boldsymbol{v}_{\mathcal{N}}^{(m+1)})$.
 12:         **if** $j > \text{Iter}_{\max}$ **then break**
 13:      $\boldsymbol{u}_{\mathcal{N}}^{(m+1)} \leftarrow \boldsymbol{u}_{\mathcal{N}}$
**Output:** A sequence of velocity profiles at corresponding time steps $\{\boldsymbol{u}_{\mathcal{N}}^{(m)}\}_{m=\ell+1}^{\ell+n_t}$.

---

Then,

$$DJ(\boldsymbol{v}; \boldsymbol{\xi}) := \lim_{\tau \to 0} \frac{J(\boldsymbol{v} + \tau\boldsymbol{\xi}) - J(\boldsymbol{v})}{\tau} = \frac{\mathrm{d}}{\mathrm{d}\tau} J(\boldsymbol{v} + \tau\boldsymbol{\xi})\Big|_{\tau=0}$$

$$= \langle DR(\boldsymbol{v})\boldsymbol{\xi}, (-\Delta)^{-1}R(\boldsymbol{v})\rangle_{\mathcal{V}', \mathcal{V}}.$$

where

$$DR(\boldsymbol{v})\boldsymbol{\xi} = \partial_t \boldsymbol{\xi} + \frac{1}{2}\big((\boldsymbol{\xi} \cdot \nabla)\boldsymbol{v} + (\boldsymbol{v} \cdot \nabla)\boldsymbol{\xi}\big) - \nu\Delta\boldsymbol{\xi}$$

is the Fréchet derivative $DR(\boldsymbol{v}): \mathcal{V} \to \mathcal{V}'$ by Lemma E.6. The Hessian is then

$$\text{Hess}\, J(\boldsymbol{v}; \boldsymbol{\xi}, \boldsymbol{\zeta}) = \big\langle DR(\boldsymbol{v})\boldsymbol{\xi}, (-\Delta)^{-1}DR(\boldsymbol{v})\boldsymbol{\zeta}\big\rangle + \big\langle \boldsymbol{\zeta} \cdot D^2 R(\boldsymbol{v})\boldsymbol{\xi}, (-\Delta)^{-1}R(\boldsymbol{v})\big\rangle.$$

Now, since $R(\boldsymbol{u}) = 0$ on $\mathcal{T}_\delta$, we have

$$\text{Hess}\, J(\boldsymbol{u}; \boldsymbol{\xi}, \boldsymbol{\xi}) = \big\langle DR(\boldsymbol{v})\boldsymbol{\xi}, (-\Delta)^{-1}DR(\boldsymbol{v})\boldsymbol{\xi}\big\rangle.$$

If we assume that $DR(\boldsymbol{u})\boldsymbol{\xi} \in \mathcal{V}'$ has its functional norm bounded above and below in $\mathcal{B}(\boldsymbol{u}; \epsilon)$, one
has for any $\boldsymbol{\xi}$

$$\|\boldsymbol{\xi}\|_{\mathcal{V}}^2 \lesssim \|DR(\boldsymbol{u})\boldsymbol{\xi}\|_{\mathcal{V}'}^2 \leq \text{Hess}\, J(\boldsymbol{u}; \boldsymbol{\xi}, \boldsymbol{\xi}).$$

Simply choosing $\epsilon$ small enough such that $\boldsymbol{v}$ is sufficiently close to $\boldsymbol{u}$ in graph norm associated
with the PDE to make the coercivity above still true for $\text{Hess}\, J(\boldsymbol{v}; \boldsymbol{\xi}, \boldsymbol{\xi})$ yields the desired local
convexity. $\qquad\square$

**Theorem E.9** (Guaranteed convergence of the fine-tuning). *In addition to the same assumptions
with Theorem 3.1, suppose Lemma E.8 holds for $\epsilon \in (0, 1)$, and a given $\boldsymbol{u}_{\mathcal{N}}$ can be embedded in
$\mathcal{B}(\boldsymbol{u}; \epsilon')$ for a $0 < \epsilon' \leq \epsilon$. Denote $\boldsymbol{u}_{\mathcal{N},j}^{(k)}$ the evaluation in Line 9 of Algorithm 2, and $j$ the iteration
of optimizer in Line 10, then the fine-tuning using the new loss function (3.9) produces a sequence
$\{\boldsymbol{u}_{\mathcal{N},j}^{(k)}\}_{j=1}^\infty \subset \mathcal{B}(\boldsymbol{u}; \epsilon')$ Furthermore, suppose that the optimizer in Line 10 of Algorithm 2 has a
learning rate converging to 0. Then, then the fine-tuning using the new loss function (3.9) produces a
sequence of evaluations converging to the best possible approximation $\boldsymbol{u}_{\mathcal{N},\infty}^{(m+1)} \in \mathcal{S}$ of $\boldsymbol{u}(t_{m+1}, \cdot)$
starting from $\boldsymbol{u}_{\mathcal{N}}^{(m)}$, in the sense that for $m = \ell, \ldots, \ell + n_t - 1$*

$$\|\boldsymbol{u}_{\mathcal{N},\infty}^{(m+1)} - \boldsymbol{u}(t_{m+1}, \cdot)\|_{\mathcal{V}} \leq \|\boldsymbol{u}_{\mathcal{N}}^{(l)} - \boldsymbol{u}(t_m, \cdot)\|_{\mathcal{V}} + c_1 n_t n^{-2}|\boldsymbol{u}(t_{m+1}, \cdot)|_2 + c_2 n_t (\Delta t)^{\gamma-1}. \quad \text{(E.21)}$$

*Proof.* For simplicity, we denote $\boldsymbol{u}_0 := \boldsymbol{u}(t_l, \cdot) \in \mathcal{V}$, and $\boldsymbol{u}_{\mathcal{N}} := \boldsymbol{u}_{\mathcal{N}}^{(m)} \in \mathcal{S}|_{t=t_m} =: \mathcal{S}$. First, by
line 10 fine-tuning algorithm, if one solves the optimization in the functional norm exactly, we have

$$\theta^* = \arg\min_\theta \delta t \big\|\overline{R}(B_\gamma \boldsymbol{u}_{\mathcal{N}}(\theta))\big\|_{\mathcal{S}'}^2 \quad \text{where } \boldsymbol{u}_{\mathcal{N}}(\theta) := \boldsymbol{u}_{\mathcal{N}} + \tilde{Q}_\theta(\boldsymbol{v}_{\mathcal{N}}) \qquad \text{(E.22)}$$

where $\overline{R}(\cdot)$ is the residual functional computed using the time derivative term from an extra-fine-step solver's result

$$\overline{R}(\boldsymbol{v}) := \boldsymbol{f} - D_t \boldsymbol{v} - (\boldsymbol{v} \cdot \nabla) \boldsymbol{v} + \nu \Delta \boldsymbol{v}$$

Unlike representing the derivative using output from the neural operator, one of the keys of our algorithm is that the error for $\partial_t \boldsymbol{u} - D_t \boldsymbol{u}_{\mathcal{N}}(\theta)$ can be explicitly estimated using the framework to develop estimates for truncation error in traditional time marching schemes for NSE. Due to the choice of the time step, this truncation error will be of higher order. Note for any $\boldsymbol{v}_{\mathcal{N}}, \tilde{Q}_\theta(\boldsymbol{v}_{\mathcal{N}}) \in \mathcal{S}$, as a result, solving (E.22) is equivalent to solve the following: denote $\boldsymbol{u}_\delta := B_\gamma(\boldsymbol{u}_{\mathcal{N}} + \delta \boldsymbol{u})$ for $\delta \boldsymbol{u} \in \mathcal{S}$

$$\min_{\delta \boldsymbol{u} \in \mathcal{S}} (\delta t)^{1/2} \left\| D_t(\boldsymbol{u}_\delta) + (\boldsymbol{u}_\delta \cdot \nabla) \boldsymbol{u}_\delta - \nu \Delta \boldsymbol{u}_\delta - \boldsymbol{f} \right\|_{\mathcal{S}'} .$$

Replacing $D_t(\cdot)$ by $\partial_t(\cdot)$ we have a truncation error term, whose error is of order $(\Delta t)^{\gamma-1}$ due to taking the time derivative:

$$\begin{aligned} & \left\| D_t(\boldsymbol{u}_\delta) + (\boldsymbol{u}_\delta \cdot \nabla) \boldsymbol{u}_\delta - \nu \Delta \boldsymbol{u}_\delta - \boldsymbol{f} \right\|_{\mathcal{S}'} \\ \leq & \left\| \partial_t \boldsymbol{u}_\delta + (\boldsymbol{u}_\delta \cdot \nabla) \boldsymbol{u}_\delta - \nu \Delta \boldsymbol{u}_\delta - \boldsymbol{f} \right\|_{\mathcal{S}'} + \left\| D_t(\boldsymbol{u}_\delta) - \partial_t \boldsymbol{u}_\delta \right\|_{\mathcal{S}'} \end{aligned}$$

We focus on estimating the first term above,

$$\min_{\delta \boldsymbol{u} \in \mathcal{S}} (\delta t)^{1/2} \left\| \partial_t \boldsymbol{u}_\delta + (\boldsymbol{u}_\delta \cdot \nabla) \boldsymbol{u}_\delta - \nu \Delta \boldsymbol{u}_\delta - \boldsymbol{f} \right\|_{\mathcal{S}'} \tag{E.23}$$

By the fact that $\| \cdot \|_{\mathcal{S}'}$ inherit the scaling law and the triangle inequality from $\| \cdot \|_{\mathcal{V}'}$, it is convex as well in this neighborhood. As a result, any gradient-based optimizer with a linearly converging step size shall converge to the minimum, achieved at $\boldsymbol{u}_{\mathcal{N},\infty}^{(m)}$, with a linear convergence rate. Moreover, $\boldsymbol{u}_{\mathcal{N},\infty}^{(m)} := \boldsymbol{u}_{\mathcal{N}}^{(m)} + (\delta \boldsymbol{u})^*$ is the (nonlinear) Galerkin projection of $\boldsymbol{u}(t_m, \cdot) \in \mathcal{V}$.

$\square$

