# OpenReview forum: "Spectral-Refiner: Accurate Fine-Tuning of Spatiotemporal Fourier Neural Operator for Turbulent Flows"
_ICLR.cc/2025/Conference — ICLR 2025 Poster_

### Official Review · Reviewer_rmBG · 2024-10-18

**Soundness:** 3
**Presentation:** 4
**Contribution:** 3
**Rating:** 8
**Confidence:** 4

**Summary:**

This paper according to me addresses limitations in current neural operator approaches for modeling turbulent flows governed by the Navier-Stokes equations. The key contributions are:
1. A new spatiotemporal adaptation (ST-FNO) of Fourier Neural Operators to enable learning maps between Bochner spaces, allowing arbitrary-length temporal predictions.
2. A novel training-fine-tuning paradigm that combines limited epochs of end-to-end training with targeted fine-tuning of a spectral convolution layer.
3. A new loss function for fine-tuning based on a functional-type a posteriori error estimator using a negative Sobolev norm, which is reliably evaluated through Parseval's identity.
4. Empirical demonstration of significant improvements in both computational efficiency and accuracy compared to existing methods.
Overall I like the paper! It's important for a lot of problems that require FNO to predict multiple temporal steps, especially in high Reynolds number ranges for NS.

**Strengths:**

1. Theoretical foundation: I like that the paper provides a rigorous mathematical analysis of discretization mismatch errors and proves the reliability of the proposed error estimator.
2. Novel architecture: The ST-FNO adaptation enables flexible spatiotemporal predictions, addressing a key limitation of existing neural operators. This is extremely useful in various timestepping problems.
3. Efficient training paradigm: The proposed approach of limited training followed by targeted fine-tuning offers a computationally efficient alternative to standard end-to-end training.
4. Strong empirical results: The method demonstrates significant improvements in accuracy and efficiency on challenging turbulent flow benchmarks.
5. Open-source implementation: The authors provide code to reproduce their results, enhancing reproducibility and potential impact.

**Weaknesses:**

1. Limited scope: The method is primarily designed for rectangular domains and uniform grids, which may limit applicability to more complex geometries.
2. Dependency on numerical solvers: The fine-tuning process relies on traditional numerical solvers for computing extra field variables, which may introduce computational overhead.
3. Assumption sensitivity: The theoretical guarantees rely on certain assumptions about the closeness of solutions, which may not always hold in practice, I'm not sure how much it does in practice.
4. Long-term stability analysis: While short-term performance is demonstrated, a more extensive investigation of long-term stability for time-dependent problems would be valuable. This could include chaos indicators or Lyapunov exponent analysis for longer time horizons, particularly for highly turbulent regimes.

**Questions:**

Just some questions:
1. Computational efficiency trade-offs: Can you provide a more detailed comparison of the computational costs of your method (including the fine-tuning phase) versus traditional CFD methods and pure neural network approaches? Specifically, how does the accuracy-to-compute-time ratio compare across these methods for different problem scales?
2. Error accumulation in multi-step predictions: For multi-step predictions, how does the error accumulate over time compared to traditional numerical methods? Is there a point at which the accuracy degrades significantly, and if so, how might this be mitigated?
3. Spectral layer fine-tuning: You mention fine-tuning only the last spectral convolution layer. Have you experimented with fine-tuning other layers or using a different architecture for this layer? How sensitive is the method to the design of this final layer?
4. Arbitrary-length temporal predictions: Can you elaborate on how your ST-FNO handles arbitrary-length temporal predictions? Specifically, how does the performance change as the prediction length increases beyond what was used in training? Are there any limitations on how far into the future the model can reliably predict?
5. Handling of boundary conditions: How does your method handle different types of boundary conditions, especially for the vorticity-streamfunction formulation? Can it adapt to changing boundary conditions without retraining?
6. Temporal super-resolution: Can your method perform temporal super-resolution, i.e., predict at a higher temporal frequency than the training data? If so, how accurate are these interpolated predictions?

---

> ### Author Response · Authors · 2024-11-27
> **Response to rmBG part 1**
>
> > Limited scope on rectangular grids and uniform grids.
>
> This is true. We want to remark that one can definitely apply the nice methods from the GNOT paper (Hao et al. ICML 2023) to couple FNO with Transformer-encoder-based NOs to generalize to general grids and geometries. Here, we chose a simple playground to present our new methods (isotropic turbulence in a periodic box, learning the energy cascade in a few epochs).
>
> ----
>
> > extra field variables introducing computational overhead
>
> If one needs to march 10000 steps to go through a given time interval, and every step needs to engage a solver, then the overhead is huge. This is the case of the traditional way of using traditional time marching solvers. Yet, if one only needs 40 snapshots, the cost is very small in this hybrid scheme. Thanks to the reviewer's question, we realized that Table 4 may lack some more direct comparisons, and in the revision, we added a time comparison to make Table 4 more informative to address this question.
>
> ----
>
> > Have you experimented with fine-tuning other layers or using a different architecture for this layer? How sensitive is the method to the design of this final layer?
>
> One of starting point and motivations of this study is that we ask the following question: *"FNO should have the capacity to achieve high accuracy because of the correspondence between the trainable weight and the coefficients for the spectral basis, why the end-to-end cannot get there?"*, and the whole discovery process of this study is to try to find a way to unlock the spectral basis's intrinsic approximation capacity. Consequently, it is actually our intention to ***NOT*** try other architectures since the presentation as a whole relies on this delicate correspondence. There are several other reasons why we did not try fine-tuning the previous layer's weights:
>   - the output's spectral expansion does not explicitly depend on the previous layer's weights.
>   - the nonlinearity from previous layers renders the optimization problem no longer convex.
>   - in the PINO paper (arXiv 2111.03794), the whole network fine-tuning for single instances was shown to have some improvement over end-to-end training, yet still cannot achieve ground truth's level of accuracy. We are technically in a different racetrack from PINO, so we decided not to copy PINO's error numbers here to make our result appear much superior.
>
> ---
>
> > Error accumulation in multi-step predictions: For multi-step predictions, how does the error accumulate over time compared to traditional numerical methods? Is there a point at which the accuracy degrades significantly
>
> If the proposed hybrid solver is applied multiple times, then stability is not guaranteed, and the error accumulates to the point that fine-tuning yields results that totally de-correlate with the ground truth. We selfishly chose not to report this due to reserving this for a future study. If you want to take a look, we can add this negative result in Table 7.
>
> ----
>
> > The theoretical guarantees rely on certain assumptions about the closeness of solutions, which may not always hold in practice, I'm not sure how much it does in practice.
>
> Unfortunately, we rely on the tools from the convergence theory of nonlinear Galerkin projection (minimizing a nonlinear residual in the $H^{-1}$ norm), whose convergence is local. Yet, in practice and in our toy models (isotropic turbulences), this "closeness" is pretty straightforward to achieve (5-15 epochs of training, no need to do 500 epochs) even for certain out-of-distribution cases, which is another reason we chose NSE as our playground to present this method.
>
> ----
>
> > A more extensive investigation of long-term stability for time-dependent problems would be valuable. This could include chaos indicators or Lyapunov exponent analysis for longer time horizons, particularly for highly turbulent regimes.
>
> We greatly appreciate this excellent suggestion by pointing out this weakness in the current presentation of the method! The long-time stability is definitely worth investigating as a future study. In fact we actually currently are investigating it by exploiting some combinations of BDF formulations and neural operators. We hope that Reviewer rmBG would not mind that we selfishly decided not to spoil it here.

---

> ### Author Response · Authors · 2024-11-27
> **Response to rmBG part 2**
>
> > Arbitrary-length temporal predictions: Can you elaborate on how your ST-FNO handles arbitrary-length temporal predictions?
>
> In our modification, the arbitrary-length temporal prediction works exactly like the spatial super-resolution in the original FNO (and its variants). Using the FNO for spatial-only tasks as an example, if the computational domain is $\Omega := (0, 2\pi)^2$ together with PBC, using the fact that channel reduction/mixing is resolution-independent, FNO can perform inference on a grid of size $64^2$ or $256^2$. Now, if we add the time dimension, say the computational domain becomes $\mathcal{T}\times \Omega := (0, 1]\times (0, 2\pi)^2$. Rolling-out approaches are natural by treating the number of snapshots as the number of channels, yet the time step sizes are now fixed. For example, one can get an output of dimension $10\times 64^2$ or $10\times 256^2$ after 10 rolling-out evaluations. Using the ST modification, one can train on a grid of size $10\times 64^2$ and inference on $40\times 256^2$. We also note that the original FNO3d uses spatiotemporal FFT, but was bounded by a global pointwise Gaussian normalization (please see the footnote on page 6), and interpolating the stored mean/std (of size $10\times 64^2$) of the normalizer from of the training data snapshots onto the inference grid (say $40\times 256^2$) does not make mathematical sense. Back to the question, in this sense, the ST modification is a natural way to do "zero-shot temporal super-resolution". However, unfortunately under the current pipeline, it cannot maintain accuracy when being applied in an autoregressive fashion, which goes beyond the horizon prepared in the training data (which goes back to the stability question). After the third application, the predicted trajectory will de-correlate significantly with the highest resolution ground truth.
>
> ----
>
> > Handling of boundary conditions: How does your method handle different types of boundary conditions, especially for the vorticity-streamfunction formulation? Can it adapt to changing boundary conditions without retraining?
>
> From the neural operator architectural level, the proposed modification cannot adapt to other boundary conditions without retraining. One has to use tricks from other work to handle non-periodic case, such as decompose $\boldsymbol{u} = \phi(\boldsymbol{x})\hat{\boldsymbol{u}}(t, \boldsymbol{x})$ for the non-slip BC case where $\phi=0$ on the non-slip boundary. We also would like note that a sufficient condition for isotropic turbulence to form is the PBC.
>
> ----
>
> > Temporal super-resolution: Can your method perform temporal super-resolution, i.e., predict at a higher temporal frequency than the training data? If so, how accurate are these interpolated predictions?
>
> Our answer to this question may be a little convoluted. We chose NSE-based data in that even for the "out-of-distribution" data (initial energy config that has higher frequency part, different Re), they can evolve into the same energy cascade. As a result, the data can then be well-represented by the model. This is also the reason why PDE-refiner, TSM, and LI all have a "burn-in" time for their data.
> Inspired by this question, and a similar request by reviewer 9ebp, we added several additional examples (together with updated codes and data in our anonymous dataset link and repo) in the Appendix D.4 on page 25-27 to report some new experiments. Back to the question, though we are fairly sure that the trained NO cannot represent higher-than-training temporal frequency under end-to-end, we find it difficult to generate such a trajectory due to energy dissipation from the viscous term. Another way to think about this question is using the (semi-)Lagrangian formulation of fluid, and the particle velocity is obtained from an exponential integrator. This integrator is usually expanded with 1 or 2 terms, which essentially shows that the time dimension can be well-represented by low-frequency components (or band-limited components).
> In this sense, our study has a philosophical take of "data are the most important, and we create a pipeline to learn the data more efficiently".

---

> ### Author Response · Authors · 2024-12-02
>
> Dear Reviewer rmBG,
>
> With the discussion ending today, we are writing to follow up on our rebuttal responses, especially several honest takes on the limitation of our methods and the reason for fine-tuning only the last spectral convolution layer. Please check on the revised manuscript uploaded on Nov 27 if possible. If there are any remaining questions needing clarification, please feel free to let us know. We are more than happy to address them before the end of the discussion period.
>
> Authors of Submission 7312

---

### Official Review · Reviewer_5tCD · 2024-11-03

**Soundness:** 2
**Presentation:** 2
**Contribution:** 2
**Rating:** 5
**Confidence:** 2

**Summary:**

This paper presents a novel learning framework that enhances the capabilities of operator-type neural networks, specifically focusing on Fourier Neural Operators (FNOs) for solving spatiotemporal Partial Differential Equations (PDEs). Recognizing the high training costs and variable accuracy of existing models, the authors introduce a spatiotemporal adaptation that allows FNOs to learn mappings between Bochner spaces, enabling arbitrary-length temporal super-resolution. The framework integrates insights from traditional numerical PDE techniques to refine the conventional end-to-end training process. For turbulent flow modeling with the Navier-Stokes Equations (NSE), the approach involves initial training of the FNO for a limited number of epochs, followed by fine-tuning a new spatiotemporal spectral convolution layer without frequency truncation. A unique fine-tuning loss function using a negative Sobolev norm, defined through a functional a posteriori error estimator based on the Parseval identity, is introduced. This loss function simplifies the optimization process, as it is convex, contrasting with the nonconvex challenges typical of end-to-end training. The proposed method significantly improves both computational efficiency and accuracy in numerical experiments on standard NSE benchmarks, outperforming traditional numerical PDE solvers and end-to-end evaluations.

**Strengths:**

The paper introduces a novel learning framework that effectively combines traditional numerical PDE techniques with operator-type neural networks, enhancing the capabilities of Fourier Neural Operators (FNOs) in solving spatiotemporal PDEs. The ability to generalize FNO variants to learn maps between Bochner spaces and perform arbitrary-length temporal super-resolution is a significant advancement for dynamic environments. By proposing a training strategy that involves limited initial epochs followed by fine-tuning, the authors address the challenges of high training costs while maintaining performance. This approach can make training more accessible and efficient.
The introduction of a fine-tuning loss function that is convex simplifies the optimization process, reducing the complications often associated with nonconvex problems in neural network training. The framework demonstrates significant improvements in computational efficiency and accuracy through numerical experiments on commonly used Navier-Stokes Equation benchmarks, providing strong empirical support for the proposed methods.

**Weaknesses:**

As with many neural network-based approaches, the interpretability of the results may be challenging.

**Questions:**

Could you elaborate on the motivation for learning maps between Bochner spaces? How does this choice influence the model’s performance and its ability to handle temporal super-resolution?

Given the model’s complexity, how interpretable are the results?

How sensitive is the model to hyperparameter choices, especially those related to the spatiotemporal spectral convolution layer? Are there guidelines for selecting optimal hyperparameters for new applications?

---

> ### Author Response · Authors · 2024-11-27
>
> > the interpretability of the results may be challenging. Given the model’s complexity, how interpretable are the results?
>
> The interpretability of our modification is actually quite good, although our original writing may have missed this nice point. Quite contrary to the traditional black-box models, our modification to the FNO and how the whole fine-tuning actually centers around our attempt to make FNO-based models more interpretable. Both the analysis and the fine-tuning rely on the fact that the weight in the last spectral convolution layer corresponds to the coefficients of the Fourier basis (3.1). In the meantime, the spectral loss function in the negative Sobolev mesh quite well between theory and practice. In theory, $H^{-1}$ norm corresponds to a weight in the frequency domain emphasizing the low-frequency part. See (3.8) for example, the higher the frequency is, the smaller the weight is. While in practice, PDE-refiner paper first discovered that the dominating part of the error (in terms of magnitude) is still the low-frequency part, and we designed a much cheaper way to address this problem without resorting to DDPM-type approach. BTW: PDE-refiner and we are on different racetracks, though; they tried to address the long-time roll-out question, and we tried to make the prediction accurate using a relatively cheap method.
>
> ----
>
> > Could you elaborate on the motivation for learning maps between Bochner spaces? How does this choice influence the model’s performance and its ability to handle temporal super-resolution?
>
> The first inspiration is that the spatiotemporal PDE's theoretical analysis is done under Bochner spaces. For Bochner spaces, e.g., $u \in C^0([0, T], H^1(\Omega))$, $u$ can be evaluated at any time $t \in [0, T]$. This trait of arbitrary evaluation in the temporal dimension of Bochner spaces is exactly the counterpart to the original "zero-shot" spatial supersolution of FNO for spatial-only problems (such as the Darcy porous medium benchmark). To see this, if we opt for the spectral basis interpretation of an FNO layer, then any basis in (3.1) can be evaluated at arbitrary point in both spatial and temporal dimensions (barring the necessity of a uniform grid for the inverse transform). For example, $e^{i\mathbf{k}\cdot\mathbf{x}}$ can be evaluated on a grid of `torch.linspace(0,1,steps=128)` as well on `torch.linspace(0,1,steps=1024)`, and this is the reason why FNO for spatial-only tasks can do super-resolution. Now, if one opts for the common neat trick of "time step $\approx$ channels" in neural operators for spatiotemporal tasks, the number of snapshots this NO accepts as input and this NO can inference are fixed (because channels are fixed once the NO's parametrization is fixed). As a result, the natural temporal super-resolution capacity enabled by space-time FFT in 3D is totally wasted. Our ST modifications remove this "time step $\approx$ channels" dependence, and thus achieve temporal super-resolution.
>
> ----
>
> > How sensitive is the model to hyperparameter choices, especially those related to the spatiotemporal spectral convolution layer? Are there guidelines for selecting optimal hyperparameters for new applications?
>
> Once the ST-FNO reaches a certain threshold of representation capacity (to well represent the energy cascade), the choice of hyper-parameters does not matter that much. This is quite counter-intuitive against the common wisdom of learning rate bounds due to SGD's convergence property. The convergence of different seeds proves this partially (see our error bar figure on the energy cascade convergence). Once entered the fine-tuning stage, the learning rate can be be pretty large (e.g., the learning rate of fine-tuning is 1, see `ex2_SFNO_finetune_fnodata.ipynb` example file as well) since the optimizing the fine-tune loss is a **convex** problem.

---

> ### Author Response · Authors · 2024-12-02
>
> Dear Reviewer 5tCD
>
> We hope this message finds you well. We are writing to follow up on our rebuttal responses, as well as on the revised manuscript uploaded on Nov 27. We would like to check whether our answers and revised paper have addressed your questions on interpretability, what our method is about, and why to set the stage using Bochner spaces. If there are any remaining questions needing clarification, please feel free to let us know. We are more than happy to address them before the end of the discussion period (which is today).
>
> Authors of Submission 7312

---

### Official Review · Reviewer_9ebp · 2024-11-04

**Soundness:** 3
**Presentation:** 3
**Contribution:** 3
**Rating:** 6
**Confidence:** 4

**Summary:**

This paper introduces the spatiotemporal adaptation technique for all FNO variants (ST-FNO), enabling them to learn mappings between Bochner spaces. The authors propose a novel strategy for training and evaluating ST-FNOs, which surpasses existing methods in both speed and accuracy. Numerical experiments on Navier-Stokes Equations (NSE) benchmarks demonstrate substantial improvements in computational efficiency and accuracy.

**Strengths:**

- The concepts of the spatiotemporal adaptation technique and the hybrid operator learning paradigm appear novel.
- The method is solidly supported by theoretical evidence.
- Overall, the paper is well-written and clear.

**Weaknesses:**

- The authors assert that their methods "yield accuracy comparable to traditional numerical methods, yet with computational resources akin to evaluating NOs." This claim would indeed be exciting if validated. However, the paper's experiments appear insufficient in the following aspects:

  - For both FNO and ST-FNO, while they may perform well on the training data distribution, they likely fail when tested with out-of-distribution data. In extreme out-of-distribution scenarios, can ST-FNO still maintain accuracy comparable to traditional numerical methods within a limited fine-tuning timeframe?

  - Table 4 compares the FLOPs of different methods, but the running times for each method are absent. I am concerned about whether the sum of inference and fine-tuning times for ST-FNO is comparable to the inference time for FNO alone.

**Questions:**

Can ST-FNO be applied to other equations and still perform effectively?

In line 65, the phrase “Similar to the traditional time marching solvers” is confusing. Did the authors intend to mean "Unlike traditional time marching solvers"?

In lines 217-218, what is the definition of $q$?

In lines 1678-1679, there is a '??'. Please correct this.

---

> ### Author Response · Authors · 2024-11-27
>
> >  while they may perform well on the training data distribution, they likely fail when tested with out-of-distribution data.
>
> This is absolutely true and we agree with the reviewer on this. We wrote the paper originally in an overly ambitious tone due to the excitement of discovering an inexpensive yet accurate fine-tuning scheme. Now, we fixed the writing true to our scientific contribution to the community at the beginning of Section 3: *"Built upon a reasonably accurate approximation, the fine-tuning of ST-FNO is proposed. It is able to yield accuracy on par with traditional numerical methods on the same time horizon, and computational resources used are comparable to the evaluation of NOs"*.
>
> ---
>
> > an ST-FNO still maintain accuracy comparable to traditional numerical methods within a limited fine-tuning timeframe?
>
> Unfortunately no, we need the evaluation of ST-FNO (without fine-tuning) to give a reasonably close "initial guess" (rel. error $10^{-2}$ in $H^{-1}$ norm, thus training is necessary) for the fine-tuning to reach $10^{-5}$ and $10^{-6}$ range. Nevertheless, due to the unique nature of NSE in the regime of isotropic turbulence, different energy distributions may eventually evolve into the same energy cascade (the reason why PDE-refiner, TSM, and LI all have a "burn-in" time for their data). We added 3 positive examples and one negative example of out-of-distribution scenarios in the Appendix D.4 of the revised manuscript. Under certain scenarios, changing the Reynolds number or changing initial energy distribution eventually evolves into the energy cascade of Kolmogorov ($-5/3$ law in kinetic energy or $-3$ law in enstrophy). In this case, the fine-tuning works well. However, when they are changed simultaneously, the fine-tuning fails to converge (the initial guess is too far away from the true solution). Please check the new experiments on page 25-27.
>
> ----
>
> > running times for each method
>
> We appreciate the suggestion by the reviewer and added the time comparison in Table 4 in a less confusing manner. The wall times are measured (instead of CUDA run time given by the `autograd.profiler`) on an RTX 4500 with a 100w power limiter (so it is occupied 100% during computation, barring some overheads of our implementation).
>
> ---
>
> > Can ST-FNO be applied to other equations and still perform effectively?
>
> For NSE-like or NSE-based systems that have energy cascades, our educated guess is Yes. For spatial-only PDE with a low-dimensional structure in the data (e.g., Darcy with log-normal permeability with a random coefficient interface), you can also train a few epochs (as opposed to 500 everyone uses) and then fine-tune. We chose to present this exciting method using NSE (transient spatiotemporal problem), because this approach makes the most sense in terms of computational cost. The parallel-in-time nature of arbitrary temporal inference, largely attributed to the spatiotemporal FFT being used, can predict the whole trajectory within the training time horizon in a single evaluation. For spatial-only PDEs, if one wants to get an accurate method (for a single instance), simply applying a traditional solver is good, which uses only a fraction of cost than engaging autograd to train an NN representation. This hybrid scheme really shines in a scenario in which you need 50 snapshots of solutions but do not want to march 10000 steps to get 50 snapshots (especially higher spatial resolution needs finer time steps bounded by CFL).
>
>
> ---
>
> > line 65, lines 217-218, lines 1678-1679
>
> We fixed these typos and inaccurate statements in the revision. You were right about line 64. For $q$, we realized that the original phrasing was confusing, so we changed it to "these snapshots learners learn an operator $G_\theta: \mathcal{X} \to \mathcal{Y}$, where the number of parameters in $G_{\theta}$ is independent of the spatial discretization size, yet ***does*** depend on the number of snapshots $\ell$" around line 218. For the ? on line 1678, we greatly appreciate the reviewer spotting this error, and we removed an unused equation reference and made the statement self-contained in Algorithm 2.

---

> > ### Comment · Reviewer_9ebp · 2024-12-02
> >
> > Thank you for your response and additional experiments. After careful consideration of the overall quality of the work, I have decided to maintain my current score.

---

> > > ### Author Response · Authors · 2024-12-02
> > >
> > > We appreciate Reviewer 9ebp for your time and consideration, as well as the suggestions for improving the paper.
> > >
> > > Authors of Submission 7312

---

### Official Review · Reviewer_UD1B · 2024-11-06

**Soundness:** 3
**Presentation:** 3
**Contribution:** 3
**Rating:** 5
**Confidence:** 3

**Summary:**

This paper introduces a new learning framework to improve operator-type neural networks for solving spatiotemporal PDEs. The proposed spatiotemporal adaptation generalizes FNO to enable temporal super-resolution. By refining traditional end-to-end training with insights from numerical PDE theory, the framework trains an FNO briefly and then fine-tunes a new spatiotemporal spectral convolution layer without frequency truncation, using a novel negative Sobolev norm loss function.

**Strengths:**

1.The ST-FNO extends FNO capabilities to handle arbitrary temporal and spatial resolutions, improving flexibility and applicability for complex PDEs like NS equations.

2.By designing the ST-FNO as a zero-shot model with arbitrary-length temporal inference, the Spectral-Refiner can adapt flexibly to varying time horizons, making it well-suited for large-scale or long-term simulations without significant retraining.

**Weaknesses:**

1. Although this paper claims that Spectral-Refiner outperforms traditional numerical methods in accuracy and computational efficiency, it does not specify how the method performs under different resolutions, time steps, and initial conditions.

2. Can the model generalize to varying conditions, such as changes in Reynolds numbers or external forces? If so, please provide correlation curves and energy spectrum curves under these conditions.

3. Model performance should not be evaluated solely based on the final frame; an error propagation curve and error distribution plot are needed to illustrate the model’s overall performance.

**Questions:**

1. The paper notes a spatial grid size of 256×256 and a prediction time interval from \( t = 4.5 \) to \( t = 5.5 \), which is quite short for meaningful evaluation. This limited time horizon may not fully showcase the model’s performance, and longer predictions should be considered for comparison. Could it surpass top-performing models such as PDERefiner, TSM, and LI in terms of prediction accuracy?

2. While this method targets turbulent modeling of Navier-Stokes equations, its applicability to other nonlinear or multiphysics PDEs is not discussed. Could Spectral-Refiner be adapted to handle other PDEs, such as heat conduction or elasticity? What modifications or enhancements would be required?

3. The paper uses the H⁻¹ Sobolev norm as a tool for error estimation, yet traditional numerical analysis often questions the suitability of non-local norms. Without access to exact solutions, is this norm sufficiently stable? How well does this error estimation perform with initial conditions that lack global smoothness?

4. This method is suggested to be more efficient and stable than physics-informed neural operators , but the comparative experiments are limited. Could more comprehensive experimental data, including comparisons with popular neural operators like DeepONet, be provided?

5. The paper notes the removal of frequency truncation during fine-tuning, which may enhance the model’s ability to capture low-frequency information. However, does this increase the risk of amplifying high-frequency noise? If so, what measures could control this issue without compromising accuracy?

6. The method involves high-order Fourier transforms and complex error estimations, potentially adding significant computational cost. In practical applications, especially those requiring real-time simulation, can this method meet timing constraints? How does its inference time compare with other methods?


7. How different are the training and testing initial conditions?

---

> ### Author Response · Authors · 2024-11-27
> **Response to UD1B part 1**
>
> >   not specify how the method performs under different resolutions, time steps, and initial conditions.
>
> Inspired by UD1B's comments, we added several experiments using different initial conditions (in terms of energy distribution, though they eventually evolve into the energy cascade). In the main body of the paper, we want to focus on delivering key messages (one should use a convex optimization problem in fine-tuning, focusing on the low-frequency part to achieve high accuracy). We chose the GT resolution to be bigger than a threshold (e.g., 64x64 is not enough to generate a "good" ground truth for $Re=1000$, the grid needs to be able to resolve the diffusion), to ensure the approximation can capture the energy cascade of the ground truth. Thus we chose $256^2$ for $Re=1000$, and $512^2$ for $Re=5000$ (newly added), and $1024^2$ for $Re=10000$ (newly added).
>
> ---
>
> > Can the model generalize to varying conditions, such as changes in Reynolds numbers or external forces? If so, please provide correlation curves and energy spectrum curves under these conditions.
>
> We added the comparison in Appendix D.4.  The generalization depends on how good the end-to-end model gives the "initial guess" for the nonlinear Galerkin projection.
>
> ---
> > The paper notes a spatial grid size of 256x256 and a prediction time interval from ( t = 4.5 ) to ( t = 5.5 ), which is quite short for meaningful evaluation. This limited time horizon may not fully showcase the model's performance, and longer predictions should be considered for comparison. Could it surpass top-performing models such as PDERefiner, TSM, and LI in terms of prediction accuracy?
>
> We agree that in the playground set up by our paper this time horizon is pretty short. Note PDE-Refiner has to train an extra noise predictor (error corrector) U-net, and each roll-out needs iterations of denoising (evaluation of this noise predictor), which is hundreds of times more expensive than our approach (e.g., 1 iteration of denoising is already more expensive than fine-tuning the last layer). We did not report the result since we didn't want PDE-refiner to look bad in this regard, as we did largely get inspired by the phenomenal observation from the PDE-refiner paper. Moreover, technically speaking, we are on a different race track here (roll-out vs spatiotemporal prediction to get all time steps at once, see Figure 1). LI is more on a different track because it learns spatial only super-resolution. We also note that in the comparison for PDE-Refiner and TSM, the "error" reported in their paper is the difference with the ground truth (GT), and yet the GT itself is a somewhat mediocre approximation to the true solution because explicit methods are used.
>
> ---
>
> > model performance should not be evaluated solely based on the final frame; an error propagation curve and error distribution plot are needed to illustrate the model's overall performance.
>
> The nice thing about spatiotemporal prediction (getting all time steps' prediction in one evaluation) is that the error is largely uniformly small across all time steps, unlike the roll-out approach in which the error accumulates more and more. We originally thought that reporting the terminal was indicative of our method's capability (as the final one has indeed the biggest error but not by much), but we were careless in this regard. Please refer to Figure 13.
>
> ---
>
> > While this method targets turbulent modeling of Navier-Stokes equations, its applicability to other nonlinear or multiphysics PDEs is not discussed. Could Spectral-Refiner be adapted to handle other PDEs, such as heat conduction or elasticity? What modifications or enhancements would be required?
>
> The whole package of the spatiotemporal modification of FNO (only a few epochs of training+finetunuing) relies heavily on learning the distribution of the spectrum (energy cascade of NSE for isotropic turbulence, also observed in the PDE-refiner paper). Only for NSE, the method works the nicest because the fine-tuning scheme costs less than the traditional scheme while achieving higher accuracy. For spatiotemporal problems such as heat conduction or elasticity (that is turbulent, not that transient), the evaluation costs less, but one cannot achieve better accuracy than traditional methods like WENO (that is used to generate the ground truth).

---

> ### Author Response · Authors · 2024-11-27
> **Response to UD1B part 2**
>
> > The method involves high-order Fourier transforms and complex error estimations, potentially adding significant computational cost. In practical applications, especially those requiring real-time simulation, can this method meet timing constraints? How does its inference time compare with other methods?
>
> As long as there is **no** autograd (backprop) on a large **nonconvex** optimization, the computational cost is negligible (e.g., versus training an extra noise corrector). The autograd in fine-tuning is convex, because only the last linear spectral conv layer is fine-tuned! We originally only reported the FLOPs (floating point operations) costs, now we added a time comparison in Table 4 using RTX A4500 (note that during inference, A4500 is not saturated, so we added a wattage limiter). By the way, we did not apply higher-order Fourier transform natively, our model still uses the FFT as all FNO variants used. We realized it using matrix product with a power of $\xi$ in frequency domain (see remark under Theorem 3.2), which is the same with FFT (linear with respect to the number of degrees of freedom barring the log factor).
>
> ---
>
> > The paper uses the $H^{-1}$ Sobolev norm as a tool for error estimation, yet traditional numerical analysis often questions the suitability of non-local norms. Without access to exact solutions, is this norm sufficiently stable? How well does this error estimation perform with initial conditions that lack global smoothness?
>
> The negative ordered Sobolev norm ***IS*** is designed for rough functions (distribution of negative order per se) or unsmooth functions. $H^{-1}$ is the closure of singular "functions" that can only be written as functionals on $C^{\infty}_c$. $L^2$ is the pivot, positive ordered Sobolev spaces, such as $H^1$, are for smoother functions, but $H^{-1}$ IS the norm for unsmooth functions (even when they can only be treated as distributions).
>
> ---
>
> > This method is suggested to be more efficient and stable than physics-informed neural operators, but the comparative experiments are limited. Could more comprehensive experimental data, including comparisons with popular neural operators like DeepONet, be provided?
>
> Unfortunately, our method needs to exploit the delicate correspondence, both in the proof of Theorem 3.1 and in the implementation, between the weights in an FNO layers with the coefficient of the spectral basis (equation 3.1). Ultimately, it is an effort to white-boxify FNO. Exploring how to apply this new fine-tuning paradigm to DeepONet will be an interesting future study.

---

> > ### Comment · Reviewer_UD1B · 2024-11-28
> >
> > Thank you to the authors for the rebuttal, which addressed some of my questions. I would like to clarify that LI is not limited to learning spatial-only super-resolution, as it is trained entirely on downsampled data. While the PDE-refiner is computationally expensive to train, its performance appears to be the best among the methods discussed. In comparison, the Spectral-refiner seems more like a trade-off, sacrificing accuracy to reduce computational cost. If offline computation time is not considered, which of the two methods demonstrates superior performance?
> >
> > It would be beneficial if the paper provided more detailed information about the training data, such as the exact number of KF flow trajectories, which I did not find in the current version. Additionally, the baseline models in the paper are somewhat limited; including more baseline comparisons a week earlier could have strengthened the results. Finally, if the Spectral-refiner were extended to 3D istropic turbulence, what challenges would arise, and are there any potential solutions to address them?

---

> ### Author Response · Authors · 2024-11-28
>
> > In comparison, the Spectral-refiner seems more like a trade-off, sacrificing accuracy to reduce computational cost.
>
> More or less yes. The Spectral-refiner is essentially a spectral method in disguise, using the FNO's spatiotemporal inference as a cheap way to get initial guesses for trajectories without the need to march thousands more steps (because traditional methods are CFL-bounded).
>
> ----
>
>  > PDE-refiner is computationally expensive to train, its performance appears to be the best among the methods discussed.
>
> If the term "performance" means the difference with the ground truth (generated by traditional numerical solvers). Then yes. For the ground truth $u\_{h}$, the residual measured under the correct functional norm at a specific $t$, $\\| R(\boldsymbol{u}\_{h}) (t)\\| \_{-1, h}$,  is around $\mathcal{O}(10^{-4})$ on $512^2$ grid and $\mathcal{O}(10^{-3})$ on $256^2$ grid. This means getting closer to the ground truth to have an error $\mathcal{O}(10^{-8})$ is pointless (because the ground truth is relatively "far" away from the true solution due to explicit scheme used for convection term). We solve a convex minimization problem directly to minimize $\\|R(\boldsymbol{u}\_{\mathrm{FNO}}) (t)\\|\_{-1, h}$, which is directly minimizing the distance with the true solution. That was why we said that we are in a different arena with LI and PDE-refiner.
>
> ---
>
> > more detailed information about the training data, such as the exact number of KF flow trajectories, which I did not find in the current version.
>
> We added more in the new revision (https://openreview.net/pdf?id=MKP1g8wU0P) after line 1188 on page 23. There are 1024 trajectories in the training data and 32 in the testing data.
>
> ---
>
> > Additionally, the baseline models in the paper are somewhat limited.
>
> We agree. Since we exploit the correspondence between the spacetime spectral convolution of FNO and the spectral methods, we focus on FNO, and it would become a totally new study for other neural operators.
>
> ---
>
> > Finally, if the Spectral-refiner were extended to 3D isotropic turbulence, what challenges would arise, and are there any potential solutions to address them?
>
> The computational cost would be the biggest challenge as DNS is used here in the fine-tuning. IMEX solvers are relatively cheap in 2D (we actually rely on the fact that marching 1 step is cheap using traditional solvers). However, IMEX grows prohibitively more expensive in 3D. Personally, I would say borrowing the wisdom from traditional spectral methods would be beneficial, such as making compromises, using LES or even RANS to formulate numerical solvers. The other big challenge is losing the streamfunction-vorticity formulation (no free-lunch way to impose the divergence-free condition), and one has to resort to newly-developed projection schemes (from Shen).  Nevertheless, the neural operator's inference would still be a cheap yet nice "initial guess" for the nonlinear Galerkin projection problem, traditional methods cannot afford implicit schemes, which solves for the nonlinear projection $u$ in $(A(u) u, v) = (f, v)$ at each time step.

---

> > ### Author Response · Authors · 2024-12-02
> >
> > Dear reviewer UD1B,
> >
> > With the discussion period coming to an end today. We are writing to follow up on our responses (and the revised paper uploaded on Nov 27). If there are any remaining questions needing clarification, especially regarding our responses above, please feel free to let us know. We are more than happy to address them promptly.
> >
> > Authors of Submission 7312

---

### Author Response · Authors · 2024-11-27
**Overall response**

We appreciate reviewers UD1B, 9ebp, 5tCD, and rmBG for the effort and time spent reviewing our paper and the suggestions for helping us to improve the paper. We have uploaded a revision and here is a summary of what is changed, and in the pdf the changes are highlighted in red color.

- ***New comparisons of out-of-distribution data***: added in Appendix D.4 on page 25 to page 27 (Table 7, Figure 10, 11, 12, 13), which include:
  - Different initial energy distribution, same Reynolds number in inference and fine-tuning.
  - Same initial energy distribution, different Reynolds numbers in inference and fine-tuning.
  - The new codes and data (Re=5000 and Re=10000 testing samples) are available in our anonymous dataset link and anonymous repo. Note that the $1024^2$ fp64 DNS takes too much space and we gave the data generation on the fly for testing in the added source codes.
  - Correlation graph with the $1024^2$ reference solution using inference and fine-tuning solutions at different resolutions, see `/examples/ex2_FNO2d_McWilliams2d_2+1d_correlation.py`
- ***Rephrasing certain claims***: we dialed back some overly ambitious claims pointed out by reviewer 9ebp, and we rewrote these claims to give an honest reflection of our scientific contribution to the operator learning community.
- ***Time comparison*** added in Table 4 (raw wall time on an RTX 4500 with a 100w power limiter), for the source codes please refer to
  - Fine-tuning of ST-FNO3d: `/examples/ex2_SFNO_McWilliams2d_ft_time_comparison.py`
  - Rolling-out type evaluation of FNO3d: `examples/ex2_FNO3d_McWilliams2d_rollout_time_comparison.py`
  - IMEX RK4 stepping: `examples/ex2_RK4_time_comparison.py`

---

### Meta-Review · Area_Chair_uiWC · 2024-12-20

**Metareview:**

The work proposes a spatiotemporal adaption to the FNO, combining end to end training with fine-tuning in a negative Sobolev norm. The work is theoretically well-founded and numerical experiments convincingly show the advantages of the method.

**Additional Comments On Reviewer Discussion:**

The authors have sufficient address reviewer concerns about the experiments and toned down some previously claims about beating traditional methods.

---

### Decision · Program_Chairs · 2025-01-22

Accept (Poster)